# Rapid gene content turnover on the germline-restricted chromosome in songbirds

Stephen A. Schlebusch [1]✉, Jakub Rídl [1,2], Manon Poignet [1], Francisco J. Ruiz-Ruano [3,4,5,10], Jiří Reif [6,7], Petr Pajer[8], Jan Pačes [2], Tomáš Albrecht [1,9], Alexander Suh [3,4,10] & Radka Reifová [1]✉

The germline-restricted chromosome (GRC) of songbirds represents a taxonomically widespread example of programmed DNA elimination. Despite its apparent indispensability, we still know very little about the GRC's genetic composition, function, and evolutionary significance. Here we assemble the GRC in two closely related species, the common and thrush nightingale. In total we identify 192 genes across the two GRCs, with many of them present in multiple copies. Interestingly, the GRC appears to be under little selective pressure, with the genetic content differing dramatically between the two species and many GRC genes appearing to be pseudogenized fragments. Only one gene, *cpeb1*, has a complete coding region in all examined individuals of the two species and shows no copy number variation. The acquisition of this gene by the GRC corresponds with the earliest estimates of the GRC origin, making it a good candidate for the functional indispensability of the GRC in songbirds.

In multicellular organisms, all cells of an individual normally contain the same genetic information. There are exceptions, however, where certain sequences are eliminated from all or some of the somatic cells during development, leaving the original genetic information to be maintained in the germ cells[1,2]. An interesting example of this programmed DNA elimination has been described in songbirds, where a whole chromosome is lost from somatic cells early on in embryo development. The aptly named germline-restricted chromosome (GRC) was described for the first time in the zebra finch (*Taeniopygia guttata*)[3], with recent studies suggesting that it likely occurs in all songbirds (order Passeriformes, suborder Oscines)[4,5]. Songbirds diverged from the rest of the birds approximately 47 mya[6] and

comprise approximately 50% of all modern bird species, making them the largest vertebrate taxonomic[7] group with obligatory programmed DNA elimination. Despite the relatively wide distribution of the GRC, we still know very little about its genetic composition, evolutionary significance, and function for birds.

Besides its exclusive presence in the germline, there is little that is consistent about this chromosome. The GRC is normally maternally inherited, but paternal inheritance has been shown to be possible[8]. It occurs in a single copy in male germ cells, which is excluded from the nucleus during meiosis, and in two copies in female germ cells[9], although again, there are exceptions[10–12]. The chromosome size varies dramatically, from the largest macrochromosome in the cell (macro-

[1]Department of Zoology, Faculty of Science, Charles University, Prague, Czech Republic. [2]Institute of Molecular Genetics, Czech Academy of Sciences, Prague, Czech Republic. [3]School of Biological Sciences, University of East Anglia, Norwich, UK. [4]Department of Organismal Biology – Systematic Biology, Evolutionary Biology Centre, Science for Life Laboratory, Uppsala University, Norbyvägen 18D, 752 36 Uppsala, Sweden. [5]Institute of Evolutionary Biology and Ecology, University of Bonn, An der Immenburg 1, 53121 Bonn, Germany. [6]Institute for Environmental Studies, Faculty of Science, Charles University, Prague, Czech Republic. [7]Department of Zoology, Faculty of Science, Palacky University, Olomouc, Czech Republic. [8]Military Health Institute, Military Medical Agency, Tychonova 1, 160 01Prague 6, San Antonio, Czech Republic. [9]Institute of Vertebrate Biology, Czech Academy of Sciences, Brno, Czech Republic. [10]Present address: Centre for Molecular Biodiversity Research, Leibniz Institute for the Analysis of Biodiversity Change, Adenauerallee 127, 53113 Bonn, Germany. ✉e-mail: stephen.schlebusch@gmail.com; radka.reifova@natur.cuni.cz

GRC) to a small microchromosome (micro-GRC), with no apparent phylogenetic pattern[4,12]. This lack of conservation is in clear contrast to the apparent ubiquity of the GRC in songbirds.

It has been hypothesised that the songbird GRC might have originated from a parasitic B chromosome[13,14]. B chromosomes are supernumerary chromosomes usually present only in a subset of individuals in the population[15]. Compared to regular chromosomes – referred to as A chromosomes – B chromosomes are not essential for an organism's survival and often spread in the population via selfish non-Mendelian mechanisms[16]. The GRC and B chromosomes show many similarities[14,17]. They are largely composed of paralogous sequences from A chromosomes[5,18], occur in various copy numbers in the cell and often show atypical behaviour during meiosis[14]. Some B chromosomes even show tissue-specific elimination[19]. However, unlike B chromosomes, the consistent presence of the GRC in all songbird species studied to date[17] suggests that the GRC is indispensable. The important functions of the GRC that prevents its loss from the songbird germline are, however, still unknown.

Part of the reason why there is still so much unknown about the genetic composition of the GRC is that this chromosome is hard to sequence effectively. Since the GRC is largely composed of recently diverged paralogous sequences from A chromosomes, it is hard to differentiate it in a sequencing library[5,20–22]. In addition, gonads are composed of both somatic and germ cells, so this chromosome is only found in a subset of testis cells and a minimal proportion of ovary cells[5]. Thus, GRC sequences are underrepresented in the sequencing libraries from these tissues. Assembled sequence information from the GRC is scarce and until recently was limited to *T. guttata*[5,8,20,21]. Thus far, analyses of tissue-specific single-nucleotide polymorphisms (SNPs) and germline/somatic coverage differences have identified 269 putative genes as well as many high copy number regions on the *T. guttata* macro-GRC[5,22]. However, the total assembled length of GRC-linked sequences is 1.24 Mbp[5] plus 468 kbp[22], which is approximately 1% of the expected 150 Mbp *T. guttata* macro-GRC. Recently, a large part of the micro-GRC was assembled in the blue tit (*Cyanistes caeruleus*). Analysis of the gene content of this chromosome revealed enrichment for functions related to the synaptonemal complex, possibly associated with the non-Mendelian inheritance of the GRC[23].

In this paper, we attempt to elucidate the importance of the GRC for songbirds and to understand the short-term evolutionary dynamics of the GRC. To do so, we sequence and assemble the somatic genomes (A chromosomes) and the GRCs from the germline genomes of two closely related songbird species, the common nightingale (*Luscinia megarhynchos*) and thrush nightingale (*L. luscinia*) and compare the GRC genetic contents. These species from the Muscicapidae family diverged approximately 1.8 mya[24] and still hybridise in a secondary contact zone[25–28]. A previous cytogenetic study demonstrated that both species possess a micro-GRC[29]. Using a novel method to identify GRC reads from germline sequencing libraries, we assemble a large fraction of the GRCs for both species. Our results show rapid gene content turnover with substantial differences not only between species but even among individuals of the same species. The majority of genes on the GRC are only partially present and presumably nonfunctional. The gene *cpeb1* is the only entire gene present in all individuals with no copy number variation. We show that this gene belongs to the oldest genes on the GRC, making it the standout candidate gene with an essential function on the GRC, which might be preventing the loss of the GRC in songbirds.

## Results
### GRC size estimation using meiotic spreads
We visualised the pachytene chromosomes in testis cells using antibodies against the lateral element of the synaptonemal complex (anti-SYCP3) and centromere (CREST; see Fig. 1). These antibodies enable the identification of the unpaired, univalent GRC, which is labelled less

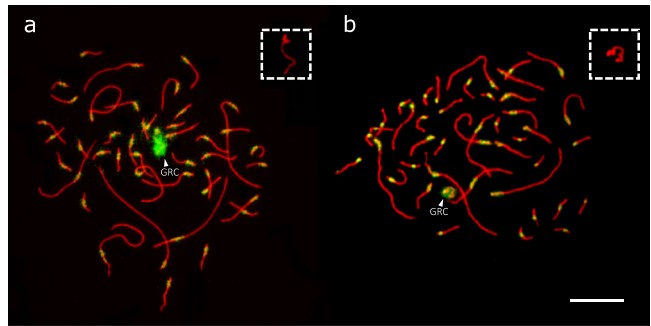

**Fig. 1 | Visualised GRC during the pachytene stage.** Pachytene chromosomes in *Luscinia megarhynchos* (**a**) and *L. luscinia* (**b**) immunostained with SYCP3 antibody against the lateral elements of the synaptonemal complex (red) and CREST antibody against centromere (green). Arrowheads indicate the GRCs. The box in the top right corner shows the GRC in more detail (1.5x magnification) without the CREST signal. The scale bar represents 10 μm. Approximately 100 nuclei in 3 individuals per species were observed before choosing these representative examples.

intensively by anti-SYCP3 compared to the A chromosomes and is covered by CREST antibody signal along its entire length[4,30]. In addition, we immunostained the eliminated GRC from the secondary spermatocytes in the form of a micronucleus (see Supplementary Fig. 1) using an antibody against histone H3 lysine 9 methylation (H3K9me)[30].

Both species had a GRC comparable in size with microchromosomes (i.e., a micro-GRC) as was described in Poignet et al.[29] Consistent with this, the GRC micronucleus for both nightingale species was much smaller than in species with a macro-GRC (see Supplementary Fig. 1; see del Priore and Pigozzi[30] and Sotelo-Muñoz, et al.[12] for visualisation of the GRC micronucleus in species with a macro-GRC).

The length of the GRC was estimated by measuring the size of the 29 largest A chromosomes, as well as the GRC, in 10 pachytene cells from 3 individuals of each species. Then, the size was compared with the assembled chromosome lengths (in bp) in collared flycatcher (*Ficedula albicollis*), a songbird species that diverged from nightingales 15 mya[31] (see Supplementary Fig. 2). Given the conservation of bird karyotypes[32], we assumed that chromosome lengths would be similar between *F. albicollis* and nightingales. Using this approach and accounting for the different chromatin state of the univalent GRC compared to the other bivalent chromosomes (see Methods)[10], we estimated the GRC size to be 8.7–9.5 Mbp for *L. megarhynchos* and 3.5–6.4 Mbp for *L. luscinia* (see Supplementary Data 1). This variation in GRC size was significant in *L. luscinia* but not *L. megarhynchos* (one way ANOVA; $p = 0.0002$; df = 29).

### Reference somatic genome assembly
A chromosome-level somatic genome assembly was created for both nightingale species. This was achieved by sequencing a female individual from each species using a combination of long-read Nanopore sequencing, standard Illumina sequencing, 10x Genomics linked-read sequencing and Omni-C chromatin conformation capture (see Supplementary Data 2 for assembly statistics). Of the BUSCO conserved set of genes[33], 94.9% were found to be complete, 1.8% were fragmented and 3.3% were missing in both genome assemblies, suggesting a high level of assembly completeness.

By aligning the assembled nightingale genomes to the publicly available *T. guttata* genome sequences (see Supplementary Fig. 3), we were able to assign the 33 largest scaffolds unambiguously to 31 chromosomes (including the Z chromosome). Chromosome 1 and chromosome 2 were both represented by two scaffolds in the genome assemblies of both *L. megarhynchos* and *L. luscinia*. This may suggest ancestral chromosomal fission events in these two chromosomes, but it is also possible that the two chromosomes were not fully assembled

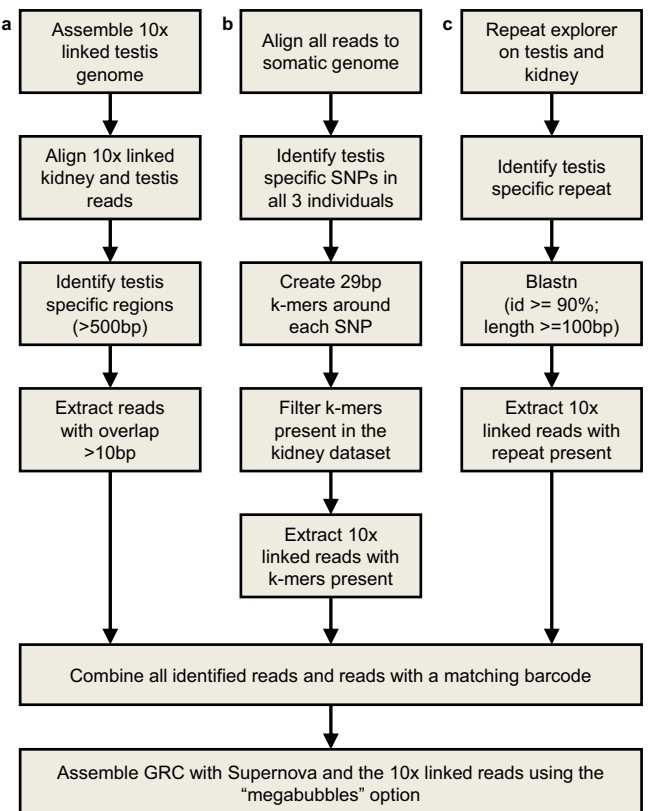

**Fig. 2 | The pipeline used to isolate and assemble the GRC reads.** A combination of three approaches (**a**–**c**) was used to identify GRC derived reads and their associated 10x linked-read barcode, before extracting all reads that shared these barcodes and using them to assemble the GRC.

**Table 1 | GRC assembly metrics for each nightingale species**

|  | *L. megarhynchos* | *L. luscinia* |
|---|---|---|
| Total length | 3.5 Mbp | 5.6 Mbp |
| Largest scaffold | 110 kbp | 370 kbp |
| Number of scaffolds | 1400 | 750 |
| Scaffold N50 | 2.8 kbp | 46 kbp |
| Scaffold L50 | 196 | 26 |
| Contig N50 | 2.5 kbp | 24 kbp |
| Contig L50 | 233 | 53 |
| %N | 3.3 | 6.4 |

and of higher quality (N50 of 46 kbp and L50 of 26; see Table 1) than the *L. megarhynchos* assembly (3.5 Mbp long with an N50 of 2.8 kbp and an L50 of 196). While the GRC assemblies were highly fragmented, their cumulative length suggests that a large proportion of each GRC was assembled (36-75% of the estimated size). This number is however probably an underestimate, as it does not take recent within-GRC duplicated sequences into account (see "Recent copy number variation within the GRC" below).

### Recent copy number variation within the GRC

The single-copy GRC coverage for each sample was estimated to be between 14-23% of the A-chromosomal coverage in *L. megarhynchos* and 15-20% in *L. luscinia*. These values were used to normalise the GRC coverage values (see "Methods" section) and calculate coverage across the GRC for the three individuals from each species. Although the coverage can vary due to sequencing and PCR bias, as well as just random chance, it can be used to identify regions of the GRC that are duplicated within the GRC but have not diverged sufficiently from their GRC paralogs to be differentiated by the assembly process[35,36]. *L. megarhynchos* showed more near-identical duplications merged in the assembly process compared to *L. luscinia*. This was reflected by the higher average normalised GRC copy number in *L. megarhynchos* (3.0x) than in *L. luscinia* (1.6x). Importantly, this mostly accounts for recent within-GRC duplications and repetitive elements on the GRC, which the genome assembler was unable to differentiate. Older within-GRC duplications that have diverged in sequence are not captured in this metric and are expected to be assembled into separate paralogous sequences.

The lower proportion of near-identical duplications in the *L. luscinia* GRC at least partially explains the higher assembly quality in this species. This is supported by the fact that the longest scaffolds from *L. luscinia* consistently had low copy number (see Fig. 3a). It is also interesting to note that there is considerable variation in copy number among individuals. This is especially noticeable in *L. megarhynchos* but is also present in the *L. luscinia* scaffolds with higher copy numbers (see Fig. 3a). This suggests that there might be substantial variation in recent within-GRC duplications between and even within species.

The normalised copy number calculation allowed for the estimation of the GRC size, accounting for near-identical duplicated sequences erroneously merged in the assembly, for each individual. Once these duplications were taken into account, the size of the GRC was 1.3–2.9x times larger than the original GRC assembly (Fig. 3b). The estimated GRC size of the individuals sequenced by 10x linked reads was 10.2 Mbp long after the correction for *L. megarhynchos* and 7.0 Mbp for *L. luscinia*. The GRC size estimate for the other two individuals of the *L. megarhynchos* were 8.8 Mbp and 12.5 Mbp while the *L. luscinia* GRC estimate was 8.8 Mbp and 11.6 Mbp. This variation is especially noticeable considering that two of the three individuals in each species are not assembled, instead using the third individual as a reference, and will therefore underestimate any sequence that does not have a homologous region in the reference GRC. The GRC size

(for example, due to some repetitive or hard to assemble region in both nightingale species). The rest of the 42 chromosomes (including the W chromosome), comprising the complete nightingale karyotype[29], were split across multiple scaffolds.

### GRC assembly

To identify GRC-derived sequences, we further sequenced and compared somatic (kidney) and germline (testis) genomes in three male individuals of each species. One individual from each species was sequenced with 10x Genomics linked-read sequencing and two individuals with standard Illumina technology. The GRC was assembled using (a) 10x linked reads that aligned in a germline-specific way to the germline genome assembly, (b) reads that contained germline-specific SNPs, and (c) reads containing germline-specific repetitive elements, which was only applicable for *L. megarhynchos*. In addition, any 10x linked reads that shared their 10x barcodes with reads selected in previous steps (see Fig. 2). This method aims to assemble not only GRC sequences that are divergent from the A chromosomes, but also GRC sequences that are similar/identical to A-chromosomal sequences, provided they have the same 10x barcode as a high confidence GRC derived read. It should be noted, however, that any large region without a high enough density of markers identifiably GRC in origin would still be missing or fragmented in the assembly. This is in addition to the normal pitfalls of an Illumina based assembly, such as sequencing biases and an inability to resolve repetitive regions[34].

Using this approach, we identified approximately 23 thousand 10x barcodes, resulting in 5.6 million read pairs to assemble the *L. megarhynchos* GRC. In comparison, only 13 thousand 10x barcodes were identified in *L. luscinia*, which resulted in 3 million read pairs. Despite having fewer reads, the *L. luscinia* GRC assembly was longer (5.6 Mbp)

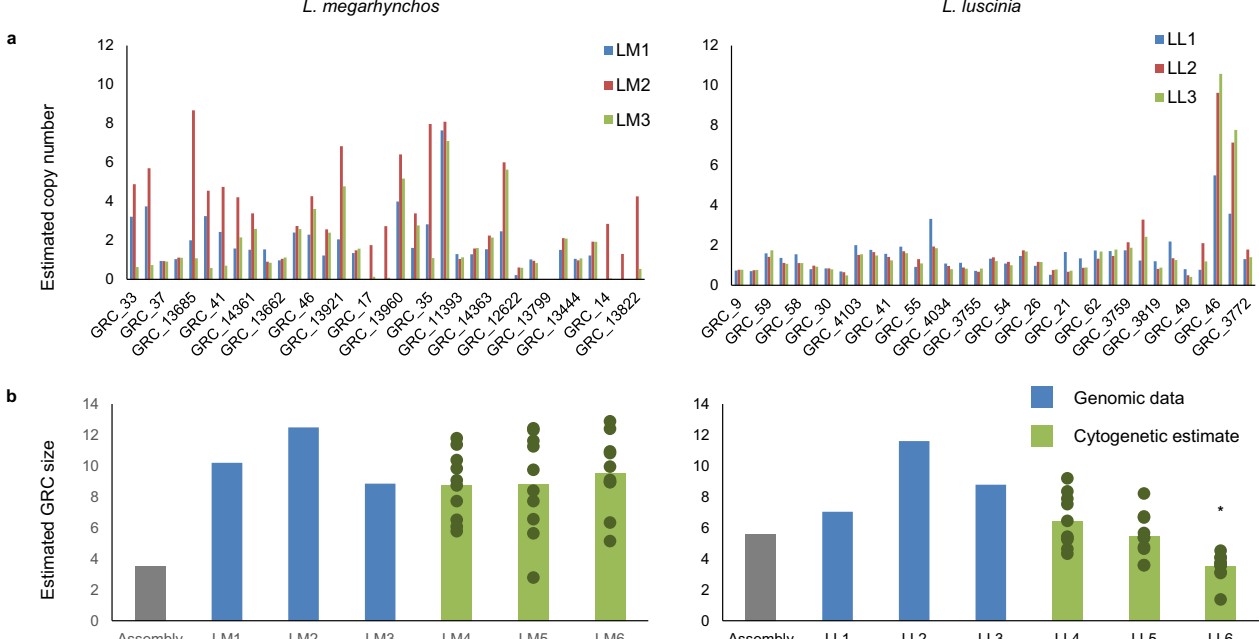

**Fig. 3 | Recent GRC copy number variation and estimated GRC size. a)** The normalised coverage from each individual was used to estimate the copy number (y-axis) for the longest scaffolds (x-axis) from the two species' GRC assemblies. Substantially more variation and higher copy numbers are observed in *L. megarhynchos* (left) than in *L. luscinia* (right). **b)** The original GRC assembly size (grey) and an estimated GRC size for each sequenced individual, taking into account the recent copy number variation on the GRC (blue), are compared to the GRC size estimates obtained from the meiotic spreads of 3 other individuals (the 10 individual measurements are shown as dark green dots with the average shown as the green bar). LM1, LM2, etc refer to the represented *L. megarhynchos* individual, while LL1, LL2, etc refer to the *L. luscinia* individual. The asterisk indicates an individual with a significantly different GRC size (One way ANOVA; $p = 0.0002$; df = 29; post hoc Tukey HSD test). Source data are provided as a Source Data file.

estimates from genomic data are similar to the GRC size estimates from the meiotic spreads (8.7–9.5 Mbp and 3.5–6.4 Mbp, respectively). Thus, although the size estimates from both the genomic and cytogenetic data should be viewed with some caution, these results do suggest that we assembled a large fraction of the GRC in both species.

### Genetic content of the GRC

The variation in GRC size between the species and individuals may also reflect different genetic content in the GRCs. To explore this possibility, we aligned the two GRC assemblies against each other as well as to the *L. megarhynchos* genome. Sequences were considered homologous if they aligned directly to the other GRC assembly (~1 Mbp in each). Additionally, to account for gaps in the assemblies which could falsely make sequences in the other assembly appear species specific, we considered sequences which originated from similar regions on the A chromosomes to be potentially homologous (see "Methods" section). This resulted in 0.5 Mbp and 1.4 Mbp of the *L. megarhynchos* and *L. luscinia* GRC being considered potentially homologous respectively. The need for this approach was confirmed with PCR amplification of three potentially homologous regions which were found to be present on the GRC in both species (see Supplementary Fig. 5f–l). Combined, our results suggest 1.6 Mbp and 2.3 Mbp of the GRC being at least potentially homologous between *L. megarhynchos* and *L. luscinia* respectively. The rest of the GRC sequences were species-specific. Once coverage is taken into account, we found that approximately one third of the *L. megarhynchos* GRC size estimate and half of the *L. luscinia* GRC size estimate is species specific. Consistent with this, the A-chromosomal origins of each species' GRC sequences are strikingly different (Fig. 4).

The *L. luscinia* GRC has a large proportion paralogous to chromosome 2, which is absent in the *L. megarhynchos* GRC assembly. Most of this chromosome 2 derived sequence comes from a single region of

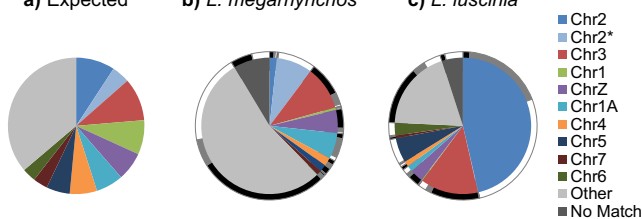

**Fig. 4 | Origin of GRC sequence. a)** The expected contribution from the A chromosomes to the GRC if each chromosome was contributing equally according to its size. **b)** The average observed contribution of A chromosomes to the *L. megarhynchos* GRC. **c)** The average observed contribution of A chromosomes to the *L. luscinia* GRC. The black, grey and white outer circles show the proportion of each sequence that is a homologous, potentially homologous or species specific compared to other species. The chromosome order is from largest to smallest, with the exception that the two scaffolds that make up chromosome 2 have been shown separately. The proportions have been corrected for coverage, which accounts for near-identical duplications collapsed in the assembly as well as assembly errors. Source data are provided as a Source Data file.

chromosome 2. This paralogous region is visible in many of the largest scaffolds from the *L. luscinia* assembly (see Supplementary Fig. 4). Despite being present in large blocks, and likely originating from a single A-to-GRC duplication, it is no longer continuous, presumably as a result of internal rearrangements and within-GRC duplications or else later A-to-GRC duplications. For example, there are two large scaffolds with chromosome 2 ancestry that are clearly within-GRC duplicates (Scaffolds 9 and 10, Supplementary Fig. 4), but because this duplication happened long enough ago, their sequences have sufficiently

diverged for the assembly process to distinguish them (see Fig. 3a). The presence of this chromosome 2 translocation in the *L. luscinia* GRC, as well as its absence in the *L. megarhynchos* GRC, was confirmed with PCR amplification of a boundary between it and GRC sequence originally derived from chromosome 6 (see Supplementary Fig. 5c-e). This confirms that the largest source of structural variation between the two GRCs is real and not a result of an assembly error.

## Divergence from A chromosomes

The proportion of mismatches between regions of each GRC and the paralogous sequence on the A chromosomes of each species was calculated to determine if the GRC sequences originated before or after the divergence of the two nightingale species (Fig. 5). If the GRC sequence was more similar to the A-chromosomal sequences of its respective species than it is to the other species' A chromosomes, it suggests that it was derived after the species diverged. By contrast, if the GRC showed similar levels of divergence from the two species' A chromosomes, it suggests that the GRC origin predates the speciation. Interestingly, a large proportion of the *L. megarhynchos* GRC appears to have originated after speciation, with over 1 Mbp of the assembly aligning better to its A-chromosomal paralogs than to the *L. luscinia* genome. In comparison, the majority of the *L. luscinia* GRC sequence appears to either predate the divergence of the species or originate shortly after their speciation, with little difference in sequence divergence between the GRC and the A-chromosomal paralogs from both species (see Fig. 5).

## Gene annotation

Genes were annotated on each of the GRC assemblies using *F. albicollis* protein-coding genes (FicAlb1.5 [https://www.ncbi.nlm.nih.gov/

datasets/genome/GCF_000247815.1/]), resulting in 192 identified genes (or fractions of genes) on the two GRCs (not including duplicates of the same gene). Of these genes, 57 were found in only one species and were located within species-specific regions of the GRC and 85 were unambiguously found on both GRCs. The remaining 50 genes were only found on one GRC, but were located within potentially homologous regions of the GRC, suggesting that their absence in one species might be caused by missing sequences in the assemblies. Indeed, PCR amplification of one such gene confirmed its presence in both species (see Supplementary Fig. 5f). The status of these genes should therefore be treated with caution. Genes were assessed for their completeness and copy number within the GRC. Notably, the vast majority of identified genes were both duplicated (with the average gene having a corrected copy number of 8.0x in *L. megarhynchos* and 5.0x in *L. luscinia*; Supplementary Data 3) and only partially present (with only 23 genes in *L. megarhynchos* and 18 genes in *L. luscinia* having more than 95% of the coding region present in the assembly; see Table 2). This observed gene fragmentation was measured after correcting for a possible lack of sequence conservation between the nightingales and *F. albicollis*, which used the percentage of the gene that was found on the A chromosomes as a baseline of expected conservation (see "Methods" section; Supplementary Data 3 contains gene-specific details).

Although some of the observed gene fragmentation is an artifact and simply the result of gaps in the assemblies (Supplementary Fig. 5g–l), several lines of evidence suggest that many genes are in fact only partially present pseudogenes on the GRC. First, many genes had premature stop codons in coding regions. Although assessing whether a specific gene copy has a premature stop codon in it was often difficult, because genes were often duplicated on the GRC and a single gene could be spread across multiple scaffolds, of all the exons identified, 16.5% (*L. megarhynchos*) to 17.5% (*L. luscinia*) had a premature stop codon. This meant that 42% of genes in *L. megarhynchos* and 35% in *L. luscinia* had at least one premature stop codon. Second, when genes were found on both GRCs, a similar percentage of the coding region was assembled (Fig. 6). The region of these genes that was present on the GRCs was also often similar; in other words, both species had the same exons present and the same exons missing (Supplementary Fig. 6). This consistency is not what would be expected if the gene fragmentation was purely the result of assembly error and instead suggests that the gene was pseudogenized in the ancestral GRC.

Of the genes with at least 95% of their coding sequence present on the GRCs, 10 were shared between the two nightingale species, from which six were previously reported on the *T. guttata* GRC[5]. Among these 10 genes, three were characterised as being homologues of endogenous retrovirus-derived proteins (*ervk* genes in Fig. 7). This includes three of the six genes that were found to be shared with *T. guttata*. The genes also include three uncharacterised genes and two homologues of Hydrocephalus-inducing proteins (*hydin* genes in Fig. 7). The two remaining genes are a zinc-finger protein (*znf239* in Fig. 7) and a homologue of Cytoplasmic Polyadenylation Element

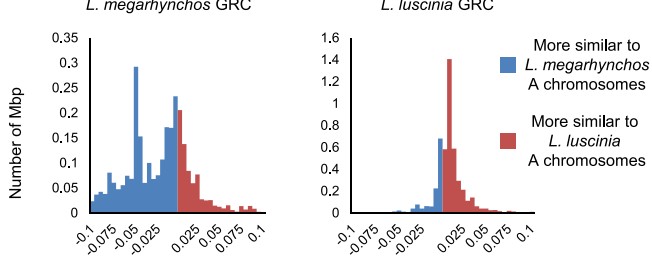

**L. megarhynchos GRC**    **L. luscinia GRC**

More similar to *L. megarhynchos* A chromosomes

More similar to *L. luscinia* A chromosomes

Number of Mbp

Difference in sequence divergence

**Fig. 5 | Histogram showing the relative divergence of GRC sequences from the A chromosomes of the two *Luscinia* species.** The x-axis shows the difference in the proportion of mismatches for a given sequence. The further away from the centre a sequence is, the larger the difference in alignment quality between the two species. Sequences that were equally divergent between the two species were removed. Sequences on the left aligned better to the *L. megarhynchos* A chromosomes (blue) while sequences on the right aligned better to the *L. luscinia* A chromosomes (red) relative to the other species. A distribution that is symmetrical suggests that overall there has been equal divergence from the two sets of A chromosomes. Source data are provided as a Source Data file.

**Table 2 | Number of GRC genes at varying levels of fragmentation identified in each nightingale species**

| Proportion of coding region found | Number of genes in *L. megarhynchos* | Number of genes in *L. luscinia* | Number of shared genes | Number of genes shared between nightingales and *T. guttata* |
|---|---|---|---|---|
| >0%[a] | 160–178 | 117–149 | 85–135 | 16–18 |
| >50% | 81–93 | 63–89 | 39–77 | 11–14 |
| >75% | 49–56 | 36–54 | 24–49 | 9–12 |
| >95% | 22–29 | 17–27 | 10–27 | 6–7 |

The lower bound of the given range is for genes identified in a species or both species with the relevant proportion present. The upper bound represents the number of genes with all potentially homologous regions. For genes shared between nightingales and *T. guttata*, the gene had to be reported as putatively on the *T. guttata* GRC[5], but the completeness was not considered.
[a]Note that genes had to have 25% of their coding region present in at least one species to be included.

Binding protein 1 (*cpeb1*). With the notable exception of *cpeb1*, these shared genes were often duplicated and differed in their copy number on the GRC, both between the species and between individuals of the same species (see Fig. 7). This means that while these genes are likely of ancient GRC linkage, they are still actively undergoing within-GRC duplication and deletion, which could suggest little selective pressure acting on them. In addition, eight of these genes had exons with premature stop codons in them in at least one of their copies, which questions their functionality. The *cpeb1* homologue, on the other hand, maintained its single copy number and full open reading frame across all exons despite having been present on the GRC for a long time, with 86 (24%) amino acid changes. It appears to have diverged from the A-chromosomal version before the common ancestor of all oscines and suboscines, early in passerine evolution (Fig. 8). The presence of

the *cpeb1* homologue on the GRC of both species was verified by PCR amplification using 2 sets of PCR primers (Supplementary Fig. 5).

## GRC gene expression

To estimate the expression of genes on the GRC, mRNA was sequenced from the adult testis and ovary of one male and female individual from each species. These data were aligned to the reference somatic genome combined with the GRC assembly of the relevant species, and while only uniquely mapped reads were used, it should be remembered that sequence similarity could affect the final expression values, either through GRC derived mRNA mapping equally well to the genome and therefore being considered ambiguous, or through A-chromosomal derived mRNA misaligning to the GRC.

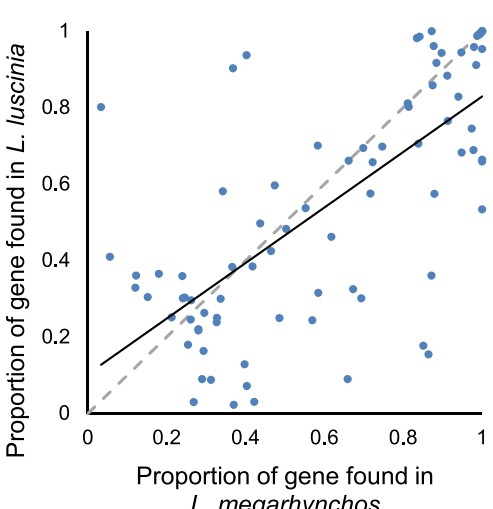

**Fig. 6 | Completeness of genes found on the GRC of both *L. megarhynchos* and *L. luscinia*.** The grey dashed line represents a 1:1 ratio. The black line represents the linear fit of the data ($R^2 = 0.49$). Source data are provided as a Source Data file.

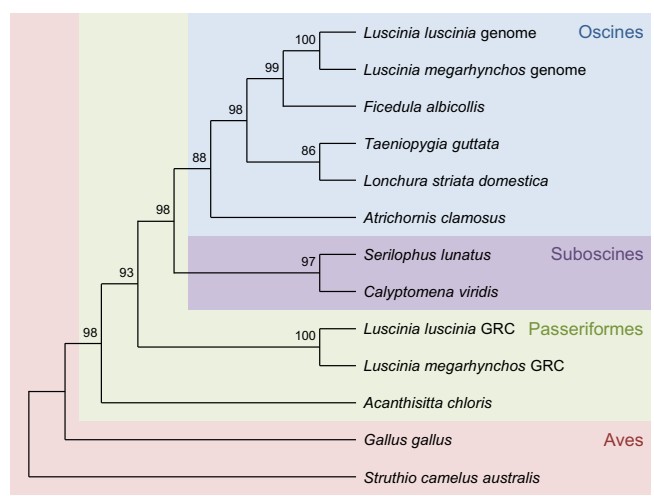

**Fig. 8 | Divergence of the *cpeb1* GRC paralog from the A chromosomal version.** The GRC paralogue diverges from the A-chromosomal version before the Oscine/Suboscine divergence during early Passeriformes evolution. Branch values represent bootstrap support.

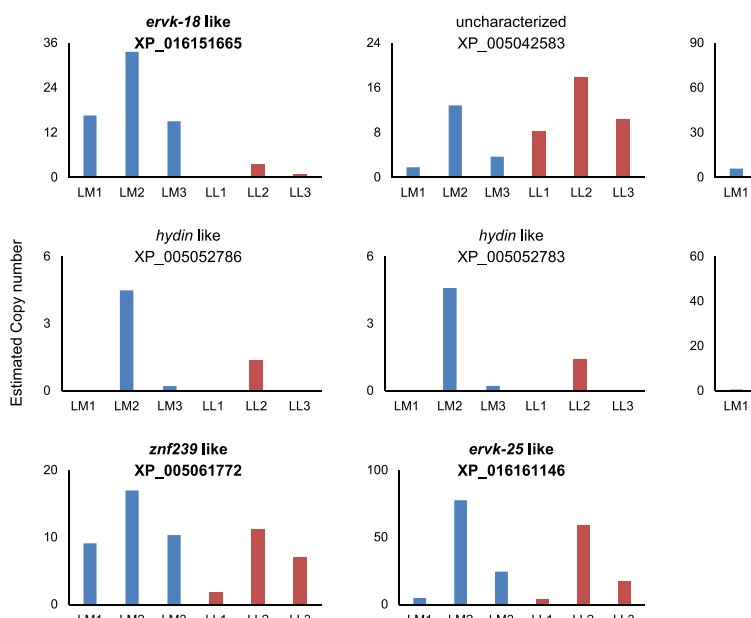

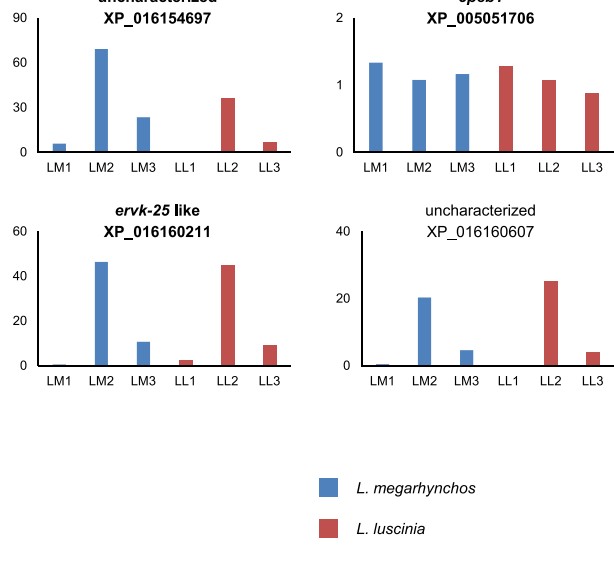

**Fig. 7 | Estimated copy number of the 10 GRC genes with at least 95% of their coding region found in both nightingale species.** Genes with bold names were also found in *T. guttata*. The copy number estimate for each individual is based off

the average normalised coverage of each scaffold that the genes were present on. Source data are provided as a Source Data file.

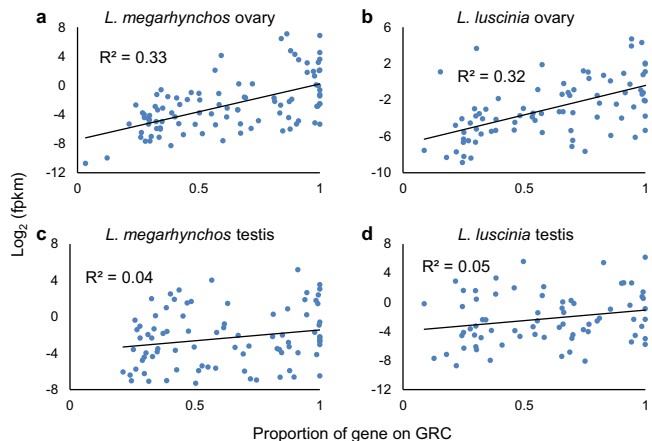

**Fig. 9 | Relationship between the completeness of a gene on the GRC and its expression level.** Expression is shown on the y-axis as the logarithmic value (base 2) of FPKM (fragments per kbp per million) which means that the decreased size of a gene has already been accounted for once. There is a correlation in the ovaries of both species (**a** and **b**), but not in the testes (**c** and **d**). Source data are provided as a Source Data file.

A total of 131 GRC genes were found to be expressed in the ovary (68% of all genes); 98 in *L. megarhynchos* (61% of its genes) and 81 in *L. luscinia* (69% of its genes). Of those genes, 48 were expressed in both species (25% of all genes). In the testis, 116 GRC genes were expressed (60% of all genes); 86 in *L. megarhynchos* (54% of its genes), 71 in *L. luscinia* (61% of its genes) and 41 in both species (21% of all genes; Supplementary Fig. 7 and Supplementary Data 3). There was also consistency in the expression levels of shared genes between the species within a tissue ($R^2 = 0.55$ in ovaries and $R^2 = 0.37$ in testes; Supplementary Fig. 8). For example, *cpeb1* was exclusively found in the ovary, where it was the sixth most expressed gene in both species. There was no correlation in expression within a species between the two tissue types however ($R^2 = −0.010$ in *L. megarhynchos* and $R^2 = 0.045$ in *L. luscinia*; Supplementary Fig. 8). Only 32 genes were expressed in both species as well as in both testes and ovaries (17% of all genes). Among them were 7 of the 10 genes which had more than 95% of their coding region present on both species' GRCs.

Interestingly, the amount of gene expression in the ovary was also correlated (exponentially) with the proportion of that gene found on the GRC (even after correcting for the smaller size of the gene fragment), with more complete genes showing higher expression than fragmented genes ($R^2 = 0.33$; see Fig. 9). This correlation was not observed in the testes ($R^2 = 0.05$). This difference between testes and ovaries may be because the GRC's maternal inheritance limits the adaption of its genes' expression in the male germline.

## Discussion

The GRC is an unusual chromosome. On the one hand, the apparent universal presence among songbirds[4] suggests that the GRC is not just a parasitic supernumerary B chromosome, as has been previously suggested[13,14], but has some important function for these birds which prevents its loss. On the other hand, our data revealed the extremely dynamic nature of this chromosome, with a lack of conservation not only between closely related sister species that diverged merely 1.8 million years ago[24], but even within species. Moreover, our results question the functionality of many of the genes identified on the GRC by observing the incompleteness of their coding regions, the presence of premature stop codons, and lower expression from shortened genes.

The GRC represents a challenge to assemble. The chromosome is only found in germ cells, which represent a small subset of cells, even when harvesting the testes or ovaries specifically. This, combined with

the fact that the GRC only occurs as a single copy in males, means that sequencing coverage of regions not duplicated within the GRC is low (about 15-25% of the A chromosome coverage in our data). Additionally, the GRC sequence is often very similar or indistinguishable from the sequence of A chromosomes[5]. For these reasons, previous attempts to identify GRC sequences, which relied on using highly repetitive GRC regions (having high germline coverage compared to somatic coverage) and germline-specific SNPs, were unable to assemble regions that have a low copy number and are not highly differentiated from the A-chromosomal sequence.

The method used in this paper to assemble the GRCs of the two nightingale species is able to assemble regions with low coverage and low divergence, as long as they have regions suitably nearby that are identifiable as GRC in origin, which can enable the classification of overlapping 10x linked barcodes. When near-identical duplications that are merged in the assembly are taken into account, our assemblies have a cumulative length that closely matches the chromosome sizes estimated from the cytogenetic visualisation, suggesting that we assembled a large proportion of the two nightingale GRCs. Together with a recently published GRC assembly in the blue tit[23], these assemblies represent the most complete GRC assemblies achieved to date, although they are still fragmented.

The GRCs of the two nightingale species are surprisingly different in sequence origin. Even after accounting for the possible gaps in our assemblies, the content of both GRCs is very different, with a conservative estimate suggesting that a third (*L. megarhynchos*) to a half (*L. luscinia*) of the GRC is species specific. This is striking, given the recent divergence of the two nightingale species (~1.8 Mya)[24] and low genetic divergence of the A chromosomes[24,26,37]. The comparison of the nightingale micro-GRC gene content with that of the *T. guttata* macro-GRC shows a similar lack of conservation. Of the 269 known GRC genes from *T. guttata*[5], only 20 were found in either of the two nightingale species (see Supplementary Fig. 6 for examples).

A stark difference between the two GRCs is the large paralogous region of chromosome 2 on the GRC of *L. luscinia*, making up approximately half of the *L. luscinia* GRC. Interestingly, analysis of the GRC and A chromosome divergence revealed that addition of this sequence to the GRC predates the divergence of the two species, suggesting the loss of the chromosome 2 paralogous region in *L. megarhynchos* rather than its recent addition in *L. luscinia*. On the other hand, we revealed that relatively large parts of the GRC in *L. megarhynchos* were added to the chromosome after the divergence of the two species. This suggests multiple frequent additions and deletions occurring on the GRC in a relatively short time span. Variation in copy number and the proportion of duplicated sequences were also high, even among individuals of the same species, suggesting that even within populations this chromosome is not well conserved. We should, however, note that our data does come from the male germline, where the chromosome is not normally heritable. Given the GRC variation is between individuals, and it does not differ dramatically between cells from a single individual, we consider it unlikely that the GRC is undergoing reorganisation specific to the testis. However, it would still be good in the future to confirm that there are no sequence differences between a heritable GRC in the ovary and a non-heritable GRC in the testis.

The striking divergence of the GRC sequence and gene content, even between such closely related species, allows for the intriguing possibility that this chromosome might be involved in speciation in songbirds. Songbirds have a higher diversification rate compared to other bird taxa and comprise more than half of all modern bird species, despite only being one of many present lineages[38,39]. We identified 57 genes inside species-specific regions on the two GRCs. This rapid gene turn-over represents a lot of opportunity for natural selection to act upon in creating reproductive isolation, both more generally among

songbirds but also specifically in nightingales, where the two species are separated by female-limited hybrid sterility and divergence of sperm morphology[25–27,40].

Most genes identified on the nightingale GRCs were only partially present. While some of this is an artifact of the assembly process, there are many compelling reasons to believe that this still represents a real phenomenon, where genes that are copied onto the GRC tend to become fragmented, non-functional pseudogenes. This is supported by the large number of premature stop codons in the assembled exons and the fact that even after accounting for length, genes had exponentially more expression the more complete they were. This pattern was also specific to the ovaries, further reducing the likelihood that this pattern and the associated gene shortening is somehow an artifact of the assembly process.

While the occasional shortening of a gene may result in a novel function, for example as a long non-coding RNA[41], it is difficult to imagine that such a process would occur simultaneously across hundreds of genes. It seems more likely that pseudogenization and an associated reduction in relative transcription (in the tissue capable of exerting a selective pressure) is the destiny of most genes that are copied onto the GRC. This pseudogenisation might be the result of little selective pressure acting on the GRC due to its presence solely in the germline, allowing for the accumulation of structural changes which would be deleterious on any other chromosome, where such mutated genes could be expressed in the tissue in which they normally act. The absence of traditional recombination on the GRC[17] would further promote the accumulation of these changes. When this is combined with the observed rapid divergence of the GRC size and genetic content between species, it suggests that large portions of the GRC are effectively non-functional and that the GRC frequently acquires (and subsequently loses) sequences from the A chromosomes. This, however, does not mean that some proportion of the GRC is not important in function. Kinsella et al.[5] found protein products for five genes on the *T. guttata* GRC, as well as signatures of selection on 10 GRC-linked genes, suggesting the functionality of at least some of the hundreds of GRC-linked genes. These genes could play important roles, for example in germline determination, oogenesis or spermatogenesis, although evolution of spermatogenesis functionality might be limited by the maternal inheritance of the GRC. While preliminary, the lack of correlation between testis gene expression and gene completeness supports this hypothesis of limited functionality in spermatogenesis, suggesting most testis GRC expression is uncoordinated and random.

In an attempt to identify conserved genes on the GRC that may be preventing its loss from the songbird germline genome, we searched for genes with a complete coding sequence that we were confident were present in both nightingale species and *T. guttata*. However, 5 of the 6 genes identified this way represent genes such as endogenous retroviral homologues and uncharacterised, or poorly characterised genes, which despite being present in all three species, also show high variation in copy number within species. It thus seems unlikely that they represent indispensable GRC sequences.

The remaining complete gene that is present in both nightingale species as well as *T. guttata* is a paralog of *cpeb1*, cytoplasmic polyadenylation element binding protein 1. In addition to the normal A-chromosomal version, this gene has a single copy on the GRC present in all 3 individuals of both species, does not feature any stop codons along its entire length, and was found to be expressed in the ovaries of both species. We estimated *cpeb1* to have diverged from the A-chromosomal version early on in passerine evolution (before the divergence of suboscines and oscines, but after the split of Acanthisittidae). This makes this gene one of the oldest genes identified on the GRC so far and suggests that the GRC might be present not only in all songbirds but all passerine birds except for a small group of Acanthisittidae. Previous analysis of this gene on the *T. guttata* GRC found that

it is under long-term purifying selection, further supporting its functionality[5]. *Cpeb1* codes for an RNA-binding protein that regulates mRNA translation during oocyte maturation and early embryonic development as well as at the post-synapse sites of neurons[42]. Maturating oocytes and early embryos are transcriptionally quiescent, with protein synthesis largely depending on post-transcriptional regulation of the stored transcripts. *Cpeb1* recognizes a specific sequence motif within the 3' UTR of the target mRNAs and modulates their expression in a spatio-temporal manner through regulating RNA cytoplasmic polyadenylation, formation of the mRNA ribonucleoprotein complex and translational repression[43–45]. In neurons, *cpeb1* contributes to synaptic plasticity and memory formation[46]. Although the function of the *cpeb1* paralog on the songbird GRC is unknown, we can speculate that it could have specialised in the oocyte-specific function, while the *cpeb1* copy on the A chromosomes holds its original functions in somatic cells. Together, these findings make *cpeb1* a tempting candidate for a functionally important gene which may be preventing the loss of the GRC from the songbird germline.

The picture emerging of the GRC is that it is a tumultuous chromosome, where large stretches of DNA can be added and subtracted rapidly, seemingly without consequence for the fitness of the organism and the probability of the chromosome being transmitted to the next generation. Once on the GRC, any sequence is liable to be duplicated on the chromosome multiple times. The pace and scale of these changes do not seem conducive to the fine scale refinement normally associated with natural selection. This, combined with the fact that genes are often fragmented, suggests that a large portion of the chromosome is non-functional, with presumably a small ancestral region harbouring genes that are driving the continued existence of the chromosome in the songbird lineage.

The seemingly contradictory picture of the GRC highlights how programmed DNA elimination can change the evolutionary landscape of genetic sequences. The fact that the GRC is eliminated from somatic cells means that there are much less pleiotropic constraints on this chromosome compared to A chromosomes, which may lead to less selection pressure acting on this chromosome. As a consequence, many genetic changes, which would have large negative consequences for an individual if they occurred on an A chromosome, are effectively silenced on the GRC.

This work represents the first comparison of GRC sequences between closely related songbirds, providing a unique insight into the evolutionary dynamics and importance of this peculiar chromosome. Our results emphasise how rapidly this chromosome evolves, with large variation being observed between the two species on almost every metric we measured, and moderate variation being observed even within each species. This contrasts starkly with the normally conserved bird karyotype and makes the GRC the fastest-evolving chromosome in the genome. We also show for the first time that many genes that are present on the GRC are present in a fragmented, presumably non-functional, state. This suggests that the chromosome is under uniquely relaxed evolutionary pressure, presumably as a result of its elimination from somatic cells. While the ubiquity of the GRC within the songbird clade does suggest an important role for the chromosome, it is still unclear what that role is. The chromosome appears to be under uniquely relaxed evolutionary pressure, presumably as a result of its elimination from somatic cells. Our results suggest that the main function of the GRC is limited to a few consequential genes, one of which seems to be *cpeb1*.

## Methods

This work was carried out in accordance with ethical animal research requirements of Poland according to Polish law (the Act On the Protection of Animals used for Scientific or Educational Purposes, 15.01.2015, item 266, implementing Directive 2010/63/EU of the European Parliament and of the European Council of 22.09.2010).

Experiments on birds were approved by the General Directorate for Environmental Protection, Poland (permission no. DZP-WG.6401.03.123.2017.dl.3).

## Nightingale sampling

Seven unrelated male individuals from each nightingale species were sampled from allopatric regions (North-Eastern Poland for *L. luscinia* and South-Western Poland for *L. megarhynchos*), three individuals for whole genome sequencing, three for cytogenetic analysis and one for RNA sequencing. In addition, one allopatric female from each species was sampled for whole genome sequencing to create a reference genome assembly as well as for RNA sequencing from their ovaries. Blood was collected from the brachial wing vein of the female individuals and was used immediately for isolation of high-molecular-weight DNA. In addition, the heart and ovaries were dissected, frozen in liquid nitrogen, and stored at −80 °C for chromatin conformation capture and RNA sequencing, respectively. From male individuals, somatic tissue (kidney) and gonadal tissue (testes) were dissected and either used for DNA isolation (which was either done immediately or after storage at −80 °C), RNA isolation (after storage at −80 °C) or preparation of meiotic spreads (testes only).

## Preparation of meiotic spreads and estimation of GRC size

Immunostained synaptonemal complexes of pachytene chromosomes were prepared as described in Poignet et al.[29] chromosomes were immunostained with rabbit polyclonal anti-SYCP3 antibody (ab15093, Abcam) (dilution 1:200) recognising the lateral elements of the synaptonemal complex, and human anticentromere serum (CREST, 15-234, Antibodies Incorporated) (dilution 1:50) binding kinetochores (see Poignet et al.[29] for details). The GRC can be recognised from other chromosomes on these slides by its relatively weaker staining by anti-SYCP3 antibody and the CREST signal which covers the whole chromosome, instead of just the centromere. The images were modified by merging the three colour channels and adjusting the contrast. Additionally, the area outside of the cell was cleaned of noise and cell debris using photoshop v.21.0.2.

The lengths of the 29 largest chromosomes and the GRCs were measured in 10 high-quality cells from 3 individuals from each species using ImageJ software (ImageJ 1.50i)[47]. The measured length of the GRC was divided by 1.5, due to a measurement discrepancy caused by its univalent nature[10], before the size was calculated using a linear regression in each cell (Supplementary Fig. 2). This used the relationship between the logarithmic values of the 29 longest chromosomes lengths and the logarithmic size in base pairs of the 29 largest chromosomes from the *F. albicollis* genome, FicAlb1.5 [https://www.ncbi.nlm.nih.gov/datasets/genome/GCF_000247815.1/] (the average $R^2$ value for a cell was 0.96 in *L. megarhynchos* and 0.98 in *L. luscinia*). The approximate size was checked against the size of the eliminated GRC micronucleus using rabbit monoclonal anti-H3K9me3 antibody (ab8898, Abcam; dilution 1:200).

## Sequencing and assembly of the two nightingales' reference somatic genomes

We sequenced the somatic genome of a female individual from each nightingale species using a combination of long-read Nanopore sequencing, standard Illumina sequencing, 10x Genomics linked-read sequencing and Omni-C chromatin conformation capture. High molecular weight DNA was isolated from 10 μl blood aliquots following a phenol-chloroform extraction protocol[48], except that 2 volumes of 80% ethanol were used for the DNA precipitation and 10 mM NaN3 was added to the TE buffer to prevent algae and moulds from growing in the final DNA solution.

Nanopore sequencing libraries were prepared using the Rapid sequencing kit (SQK-RAD004) and the Ligation sequencing kit (SQK-LSK108) (Oxford Nanopore Technologies) according to the manufacturer's instructions. The libraries were sequenced on MinION and GridION instruments using FLO-MIN106 flowcells and standard parameters. The standard Illumina sequencing was done at GeneCore (EMBL, Heidelberg, Germany), where 2 × 250 bp paired-end libraries were prepared and sequenced using the HiSeq 2500 (Illumina). For the 10x Genomics linked-read sequencing, the DNA was sent to SEQme (Dobris, Czech Republic) for library construction and 2 × 150 bp paired-end sequencing using the NovaSeq 6000 (Illumina).

Sequencing adaptors were trimmed from the Nanopore reads using Porechop v0.2.4 (https://github.com/rrwick/Porechop). Trimmomatic v0.39[49] was used for quality and adaptor trimming of the Illumina reads. The 10x linked reads were processed by the Long Ranger v2.2.2 basic pipeline (10x Genomics).

The trimmed Nanopore reads were assembled using Flye v2.7.1[50]. Minimap2 v2.17[51] was used to map the trimmed Nanopore reads on the resulting draft genome sequence followed by error correction with Nanopolish v0.13.11[52]. The genomes were further error-corrected three times using the Illumina short reads and Pilon v1.23[53] after they were mapped with BWA v0.7.17[54]. To correct erroneous hybrid scaffolds, the 10x linked reads were used with the Tigmint v1.1.2 pipeline[55]. The 10x reads were also used to further scaffold the genome sequences using Arcs v1.1.1[56] and LINKS v1.8.6[57].

To scaffold the genome assemblies to chromosome level, the deep-frozen heart sample was shipped to the Dovetail Genomics, LLC (CA, USA) for Omni-C chromatin conformation capture sequencing. The assembly was then improved using their HiRise software v0.75.

The completeness of the assembled genome was assessed with BUSCO v4.0.6[33] using the 'vertebrata_odt10' dataset. The assembled scaffolds were also aligned to the *T. guttata* genome (GenBank assembly accession: GCA_003957565.2) using Minimap2 v2.17[51] to identify the homologous chromosomes and Circos v0.69-9[58] for visualisation (see Supplementary Fig. 3).

## Sequencing of somatic and germline genomes from male individuals

We sequenced DNA from somatic (kidney) and gonadal (testis) tissues from three individuals of each species. One individual from each species had DNA from both tissues sequenced using 10x Genomics linked reads[59] Illumina sequencing technology, while the other two individuals had DNA samples sequenced with standard paired-end Illumina sequencing.

High molecular weight DNA was extracted for 10x linked-read sequencing from the frozen testis and kidney samples using a phenol-chloroform protocol[48] and sent to the company SEQme (Dobris, Czech Republic) for library construction and 2 × 150 bp paired-end sequencing using the NovaSeq 6000 (Illumina). For the standard Illumina sequencing, DNA was extracted from frozen tissue samples using MagAttract HMW DNA Kit (Qiagen) and sent to the Institute of Applied Biotechnologies (Prague, Czech Republic) where the sequencing libraries were prepared using NebNext Ultra II DNA Kit (New England Biolabs) and sequenced with the NovaSeq 6000 (Illuimna) using 2 × 150 bp paired-end mode.

Testis samples were sequenced to a higher depth (105-150x) than the kidney samples (45-120x) to ensure sufficient coverage over the GRC (see Supplementary Data 4).

## Identification and assembly of GRC reads

Linked 10x reads that originated from the GRC were identified using the following methods before being assembled using Supernova[60] and the "megabubbles" option (method visualised in Fig. 2):

(a) To identify testis-specific sequences, the testis genome was assembled using the 10x linked reads and Supernova[60]. The 10x reads from the testes and kidneys were then processed using Long Ranger v2.2.2, trimmed and checked for adaptors using Trimmomatic v0.39[49],

before having duplicates removed using FastUniq v1.1[61]. They were then aligned to their respective germline genome using bwa v0.7.17[54]. Regions of the testis genome assembly that were at least 500 bp long and were fully covered by reads from the testis dataset while having no reads align to them from the somatic dataset were identified using Samtools v1.14[62]. Reads from testis samples that overlapped these regions by at least 10 bp were used to identify 10x barcodes and their associated reads as originating from the GRC.

(b) To identify testis-specific SNPs, sequencing reads from 10x libraries were processed using Long Ranger v2.2.2 before all reads were trimmed and checked for adaptors using Trimmomatic v0.39[49] and having duplicates removed using FastUniq v1.1[61]. These reads were then aligned to the reference somatic genomes from their respective species. SNP variants were identified using GATK v4.1.7.0[63]. If a variant was present in all three testis samples from a species, but in none of the kidney samples from that species, it was considered to be a GRC variant. These SNP variants were used to create 29 bp kmer sequences (i.e. each variant resulted in 29 kmers). Any kmer that was present in the 10x kidney reads was removed and the remaining kmers were used to identify reads from the 10x testis dataset that had a matching sequence. The barcodes from these reads were used to identify any associated reads.

(c) A sample of 100 000 reads from each of the 10x datasets was used to identify repetitive elements that might be unique to the GRC of each species using RepeatExplorer[64]. While no such repeats were found in L. luscinia, a candidate repeat was found in the testis dataset of L. megarhynchos. This result was confirmed using all the 10x reads from L. megarhynchos and Blastn v2.10.0[65], with a word size of 8, an e-value of 1e-5, and a max hsps of 1. Any read that matched with at least 100 bp and greater than 90% identity to the repetitive element was selected. Once again, all reads with the associated 10x barcodes were designated as having originated from the GRC.

## Coverage of GRC scaffolds

In order to identify scaffolds that represent near-identical duplicates and/or erroneous sequences in the GRC assembly, the assembled GRC sequence for each species was combined with the corresponding reference somatic genome assembly. The sequencing reads from both tissue types for all three individuals were aligned to the combined genome and GRC assembly using bwa v0.7.17[54] for each species (see Supplementary Fig. 9). For each individual, regions of the GRC which had zero read coverage from the kidney dataset were identified. The modal testis coverage value of these regions for each individual was used as an estimate of the expected coverage for single-copy GRC regions. The ratio of the expected GRC coverage to the modal genomic coverage was used as a proxy for the "expected" GRC coverage in the kidney samples. These expected coverage values were used to normalise the observed coverage for each sample. The coverage of the kidney sample was then subtracted from the coverage of the testis sample to control for A-chromosomal reads misaligning to the GRC sequence. Finally, the average GRC coverage was calculated across 1 kbp windows along the two GRCs, with a minimum value of zero.

## GRC scaffold origin and conservation

Given that GRC sequences appear to originate from A chromosomes, the GRC scaffolds were aligned to the reference L. megarhynchos and L. luscinia somatic genomes using Minimap2[51]. Additionally, the two GRCs were also aligned to each other using Minimap2. The A-chromosomal origin of the GRC sequences were determined by whichever T. guttata chromosome corresponded to the top hit in the L. megarhynchos genome.

In order to identify regions with potential shared homology, aligned regions within 10 kbp of each other were merged using

Bedtools merge v2.27.1[66]. Regions with alignments from both species were then labelled as potentially homologous.

## Gene annotation

The first frame of the F. albicollis transcriptome coding sequences (v FicAlb1.5 [https://www.ncbi.nlm.nih.gov/datasets/genome/GCF_000247815.1/]) were aligned independently to the two assembled GRCs using Tblastx v2.10.0[65] and an e-value cut-off of 1e-6. Overlapping alignments on the same strand of the GRC were merged. The resultant regions were aligned back to the F. albicollis coding sequences and the top hit in the positive strand was selected to identify which gene (and which portion of the gene) the exon represented.

When calculating what proportion of a gene was present on the GRC, the estimated proportion of that gene which was found on the GRC was normalised by dividing it by the proportion of the gene that was found in the L. megarhynchos genome. This was done in order to account for a possible lack of conservation between the two species. The normalisation was limited by not allowing the denominator to be smaller than 0.75. In other words, if 80% of a gene was found in the genome, and 80% was found on the GRC, that gene was treated as being 100% present on the GRC, but if 5% of a gene was found on the GRC and 5% in the genome, that gene was only treated as being 6.7% present in the genome. The cut-off of 0.75 is sufficient to fully correct for 2/3rds of all genes identified on the genome. This correction resulted in an average increase in the measured proportion of the found gene of 8.5% in L. megarhynchos and 8.9% in L. luscinia. Only genes with at least 25% of their coding region present in one of the two species were considered in order to reduce the risk of short sequences misidentifying homologous proteins.

The open reading frame of each exon was determined using ORFfinder v0.4.3 (https://www.ncbi.nlm.nih.gov/orffinder/). Because the exons were annotated with F. albicollis coding sequences, no untranslated regions (UTRs) are expected in the annotated exons; therefore, an exon was deemed to have a premature stop codon if no open frame measured 95% of its length.

## RNA sequencing and analysis

Total RNA was extracted from an ovary from the female used for the somatic reference genome and the testis of a male individual from each species using TRIzol™ reagent (ThermoFisher Scientific). Of the total RNA, 9 μl was treated with TURBO™ DNase (Ambion) cleaned using an RNA Clean and Concentrator kit (Zymo Research). The samples were then sent to GeneCore (EMBL, Heidelberg, Germany) for RNA-seq library construction using the NEB Ultra II stranded mRNA kit (New England Biolabs), including Oligo dT magnetic beads for mRNA enrichment. These libraries were sequenced to result in 150 bp single-end reads using HiSeq 2500 (Illumina). RNA reads were trimmed and had adaptors removed using Trimmomatic v0.39[49]. The trimmed reads were aligned to the reference somatic genome combined with the GRC assembly of the corresponding species using STAR v2.7.9a[67]. Only reads that mapped uniquely to the GRC exons were considered.

## cpeb1 Evolution

The cpeb1 gene sequence was determined from the Tblastx results with the XP_005051706.1 transcript from the two GRCs and their respective genomes. The GRC sequence was used to identify homologous sequences in related species using the 'nr' database on NCBI and Tblastx. For each species, the top hit was selected. These species included: Acanthisitta chloris, Atrichornis clamosus, Calyptomena viridis, Corvus cornix cornix, Gallus gallus, Lonchura striata domestica, Sapayoa aenigma, Serilophus lunatus, Serinus canaria, Struthio camelus australis, T. guttata and Tyrannus savana. The sequences were aligned using ClustalW[68] and a maximum likelihood tree drawn with MegaX[68].

## PCR validation

To validate the assembly of GRC sequences, we designed two PCR primer pairs targeting the GRC copy of the *cpeb1* gene and the *L. luscinia* GRC boundary between the chromosome 2 translocation and the chromosome 6 translocation (see Supplementary Data 5). To test whether species-specific gaps in homologous regions represented real genetic differences, primers were designed for three additional genes. Two of which were partially present in both species, but to varying degrees and the other was only found in *L. megarhynchos*. Each 50-μL PCR mixture contained 0.2 mM dNTPs (Sigma-Aldrich), 0.2 μM primers (Generi Biotech), 2.5 U of JumpStart™ Taq DNA Polymerase (Sigma-Aldrich) with corresponding PCR buffer and 20–30 ng of template DNA. For each forward/reverse primer pair we prepared five PCR reactions containing *L. megarhynchos* testis DNA, *L. megarhynchos* kidney DNA, *L. luscinia* testis DNA, *L. luscinia* kidney DNA and a no template control. The reaction conditions were as follows: 94 °C for 60 s, 30 cycles of 94 °C for 30 s, 50 °C – 59 °C for 30 s, and 72 °C for 60 s with a final extension at 72 °C for 3 min. PCR products were visualised on 1% agarose gel with ethidium bromide staining (see Supplementary Fig. 5).

## Reporting summary

Further information on research design is available in the Nature Portfolio Reporting Summary linked to this article.

## Data availability

The novel whole genome Illumina sequencing data generated in this study have been deposited in the NCBI's SRA database under the BioProject accession code PRJNA808609. The GRC assemblies have been uploaded onto Figshare (doi: 10.6084/m9.figshare.19161545) and NCBI under the BioProject accession code PRJNA808609 (accessions GCA_030412455.1 and GCA_030412475.1). The assembled reference somatic genomes and their corresponding raw data have also been deposited in the to NCBI database under the Bio-Project accession codes PRJNA810511 and PRJNA810515. RNAseq data have been deposited onto NCBI via GEO with the accession code GSE215907. The FicAlb1.5 genomic data used in this study are available in the NCBI database under accession code GCF_000247815.1.and the bTaeGut1.4.pri genomic data is found under the accession code GCA_003957565.2. Source data are provided with this paper.

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

## Acknowledgements

This research was funded by the Czech Science Foundation (grant 20-23794S to R.R., T.A. and J.P., grant 23-07287S to R.R and T.A.), the Charles University grant PRIMUS/19/SCI/008 to R.R., the Grant Agency of Charles University (grant 1169420 to M.P.) and the European Research Council (101002158 GermlineChrom to A.S.). F.J.R.R. was supported by a postdoctoral fellowship from Sven och Lilly Lawskis fond and a Marie Curie Individual Fellowship (875732). The mRNA Illumina sequencing was performed in collaboration with Vladimir Benes and his GeneCore facility team (EMBL, Heidelberg, Germany). The whole genome Nanopore sequencing was supported by the Ministry of Defence of the Czech Republic through a long-term organisation development plan 907930101413. Computational analysis was mostly done using the Institute of Molecular Genetics (Czech Academy of Sciences, Prague, Czech Republic) computers. Part of this work was carried out with the support of ELIXIR CZ Research Infrastructure (ID LM2023055, MEYS CR).

## Author contributions

The project was conceptualised by R.R., S.A.S., A.S., T.A. and J.P.; Samples were collected by J.Re., T.A., R.R. and M.P.; Cytogenetic analysis was performed by M.P.; DNA and RNA extraction was done by J.Rí. PCR primers were designed by F.R-R. and S.A.S. and PCR performed by J.Rí.; Nanopore sequencing was done by P.P and the somatic reference genome assembly was done by J.Rí.; Other Bioinformatic analyses were done by S.A.S. with consultation from A.S. and F.R-R; Manuscript was written by S.A.S. and R.R. and all authors contributed to text editing.

## Competing interests

The authors declare no competing interests.
