## [Peer Review File · Nature Communications]

Rapid gene content turnover on the germline-restricted chromosome in songbirdsReviewers' Comments:

Reviewer #1:

Remarks to the Author:

This study by Schlebusch makes a more complete germline restricted chromosome (GRC) assembly of two closely related species, the common and thrush nightingale. The authors have produced two of the most complete assemblies of the GRC to date, although still highly fragmented relative to autosome assemblies. Nevertheless, this new assembly helps answer questions around this enigmatic chromosome in songbirds. Tissue was sampled from three individuals of each species. The authors use three approaches based around 10x barcoding technology to identify and assemble GRC-specific sequence. They show that GRC gene content differs dramatically between the two species and even within individuals of the same species. Only 4% of GRC genes had their entire coding region present, while the rest were in pseudogenized fragments, suggesting that the GRC is under little selective pressure. The authors identify only one gene, *cpeb1*, which is present in all individuals sampled without copy number variation and contains the complete coding region. Based on comparison with the A-chromosome paralogue in various avian species, they find that GRC *cpeb1* originated before the oscine/suboscine divergence. Given that this corresponds to the earliest estimates of the origin of the GRC, *cpeb1* is likely essential and helping to maintain the GRC in songbirds.

Overall, we are excited by the study, which presents an amazing amount of GRC reorganization between closely related species and between individuals within a species. The only other example of such reorganization we can think of is the immunoglobulin genes, but even then, here the GRC reorganization is an entire chromosome. Because of such a remarkable claim, we think the authors need to alert readers of potential technical limitations that could contribute to a higher than expected chromosome reorganization between and within species. No doubt this is a technically challenging chromosome to sequence and assemble to begin with.

First, the reference somatic assembly is done with short reads, using the 10X genomics protocol. The authors should mention the fragmented limitations of such assemblies, as shown in Rhie et al 2021 Nature for the Vertebrate Genomes Project (VGP). Second, the authors selected the reads for the GRC assembly by subtraction of those that did not match the somatic reference chromosomes, using three parallel approaches. This unmapped GRC reads by definition are going to result in a more fragmented GRC assembly, because the testis GRC reads that are identical to the somatic chromosomes will also be removed, including proportions of genes. This caveat should be mentioned.

It is hard to imagine how can over 90% of the genes loss parts of coding sequences in the GRC. This would have to be due to a highly recombinant process that breaks up and reassembles pieces of somatic DNA into the GRC. An alternative interpretation is that many of the protein coding sequences are missing due to them being identical to the somatic chromosomes and thus subtracted out before assembly of the GRC.

I think the evidence for different size GRCs between species and individuals is more convincing. But I wonder if another biological explanation could be at play, different developmental stages within an individual. As the primordial germ cells (PGCs) mature, maybe their GRC chromosome is undergoing different stages of germline chromosome reorganization, and what the authors are catching are cells in gonads that are in different stages of reorganization between and within species. If this seems plausible, it should be mentioned as a possible alternative explanation to consider.

So overall, an amazing story, but with some alternative technical and biological interpretations that need to be considered and mentioned.

Specific comments:

Given that GRC size estimates from the genomic data indicate significant within species variation

(Figure 3), we wonder if the authors could identify any similar size variation from the chromosome spreads as well?

The descriptions of the GRC assembly in the main text results and methods are confusing. The supplementary Figure 3 version of it is much more understandable. I recommend moving this figure to the main text, and explain the assemblies based on it. For example, the main text has apparent Steps 1), 2), and 3), without any statement that these refer to the pipelines of the left, middle, and right columns of Supplementary Figure 3. We had to figure that out after several readings. Does Somatic genome in the middle column mean the kidney? For the very last step, was Supernova used to assemble these apparent GRC reads? This final assembly step does not seem to be described in the methods?

Line 137. The copy number and coverage description needs to be explained better. "Collapsed repeats" is usually the term for artificial collapsing of duplications, shown with increased coverage of raw reads. "Expanded repeats" refers to the calculation that the authors made for estimating to true size of the chromosome if repeats were expanded. But the authors did not actually expand (lengthen) and reassemble the repeats? They give the impression that this was done.

Line 246. I wouldn't call these GRC assemblies high-quality yet, despite them being the most complete thus far. There are still draft quality assemblies; they can say the most complete draft-quality assemblies. High-quality GRC assemblies will probably require using Pacbio HiFi reads and more advances in technology to isolate the GRC or PGCs from other cells.

Line 269. The Prum et al 2015 bird tree has been shown to be mostly a protein-coding tree that does not reflect the rest of the genome (Jarvis et al 2014; Reddy et al 2017). I think citing Jarvis et al and Reddy et al for the relationship of songbirds with other species in the tree would be more appropriate.

lines 301 and 302: There is quite a bit known regarding *cpeb1* in the literature. Can the authors include a little more discussion of that? But, the discussion does have redundancy in it, and they can shorten that redundancy too.

line 373: Do the authors have any concern with not using a PCR-free DNA-seq kit to reduce coverage biases? Please discuss potential PCR and coverage bias.

lines 390-399: Why is a different reference genome used in method 1 and 2? Was there something to be gained?

lines 414-420: This paragraph is unclear and can be rephrased.

Line 439. Blastn is not a good tool for genome alignments. Blastn makes partial alignments. Cactus, Dialign, ClustalW don't delete in between unaligned regions, and thus one gets a more complete alignment.

Line 458. Here the authors will need to have a public accession number of genebank deposited assemblies.

Figure 2 legend needs more information in order for readers to follow what is happening on the Y and X axis.

Figure 4. Are the chromosome numbers the same as the zebra finch? Further, which zebra finch assembly was used for assigning chromosomes? The Rhie et al 2021 zebra finch assembly is the most updated GCF_003957565.2 for chromosomes. The version used needs to be mentioned and cited.

Reviewer #2:
Remarks to the Author:
Dear Dr. Jones:

Schlebusch et al. in this submitted ms investigated two recently diverged songbird species, common and thrush nightingale, for their germline restricted chromosome (GRC) sequences, and compared them between each other, and vs. their regular chromosome (A) sequences for their origins. They extracted the GRC sequences either based on their absence in somatic genome assemblies, or derived from the reads that have diagnostic SNPs or repeats that are specific to the germline. Due to the intrinsic limitation of the sequencing method (Illumina or 10x) and the repetitive nature of GRCs, the assembly is fragmented; but their analyses indicated that there can be large copy number variations of GRC sequences between individuals, and also between species. Although of recent divergence, GRCs of the two species seem to share very few genes and also underwent independent sequence amplification. Few genes annotated in both species' GRCs seem to be intact, with only one gene, CPEB1 that is inferred to be shared by both. There are many recent studies elucidating the origin and evolution of GRCs in songbirds (and absent in other bird species), started from the first genomic study of GRC in zebra finch (Kinsella et al. Nat. Comm.) to more recently, a more systematic inspection of its existence in 16 songbird species (Torgasheva et al. PNAS). I think to gain more genomic insights into GRCs of more species is important for our understanding of their origin and function, if any. However, I have to say, the new insights from this paper, compared to previous work, are not significant enough to merit a publication on Nat. Comm. My major comments are as follows:

1. One major conclusion of this work is many GRC-derived sequences contain duplications, inferred only from normalized read coverage data. However, read coverage can be notoriously variable due to the different GC or repeat content from Illumina reads, and this can be indicated by the large variation observed between different scaffolds from GRCs shown in Figure 2. I did notice that, there are higher levels of coverage variations between *L. megarhynchos*' individuals than those of *L. Luscinia*, however additional methods (e.g., check if there are actual reads spanning the duplicated sequences' boundaries, or violations of mate-pair relationships, or any evidence from the assembled sequences) are needed to make sure the duplicated regions are real. Before that, I would be very cautious to conclude, e.g., at line 141, that the current assembly size is already close to the estimate from meiotic spreads. As both are quite inaccurate that the actual size of the chromosome cannot be accurately scaled by their length, as highly heterochromatic chromosomes tend to be more compact, i.e., shorter than its scaled length.
2. The paper reported that at line 183 "585 genes were identified in *L. megarhynchos* GRC, and 406 in another species". However, it later turned out that "vast majority of these genes are duplicated and only partially present". So this is quite misleading, as the result seems to be these numbers of genes only means gene fragment, and it is unclear how many of these fragments present on the GRCs actually have an intact open reading frame, or at least have any evidence of transcription. Therefore, it is difficult for the reader to assess whether the GRCs are functional or completely a parasitic element. The authors should provide information regarding the coding potential of these genes as well as the transcription evidence.
3. Figure 5 is very unclear to me: in the main text, please introduce what is the hypothesis for using this mismatch pattern to determine when GRC originated regarding the two species divergence time. There are no legends in Figure 5, which only shows two density plot of distribution. It is also unclear how the authors estimated "1Mb of the assembly align better...". I assume that the authors hypothesis sequences that are more similar to one species' genome could have originated after this species diverged from the other. But how is this pattern relate to Figure 4? As if the GRC sequence originated after the speciation, at least this sequence should have a different A chromosome origin that is absent in the other species' GRCs, is that so? And how many sequences are like that?
4. It is also unclear that the gene fragment on the GRC is caused by the assembly gap or it is fully assembled, but the sequence open reading frame has been disrupted.
5. The discussion about CPEB1 is too short, at least this gene I considered as one major finding of the work, that it seems to be originated at the base of songbirds and could potentially explain the

existence of GRC. In the text, the author wrote "CPEB1 is known to play a role in transcript modification during oocyte maturation...this makes it the best candidate for preventing the loss of the GRC from germline". If I understand it correctly, now songbird germline contain an extra copy of CPEB1 than other birds, and it is also unclear whether the GRC-linked CPEB1 actually perform the same function as the copy on the A chromosome; and even if it indeed has the same function, how would this extra copy matter to the "oocyte maturation"? These factors all need to be discussed.

6. There should also be introduction/discussion about the distinction and similarity between the GRC and B chromosomes, both of which are extra non-mendelian chromosome elements in addition to the A chromosomes, and carry mostly repetitive sequences and few functional genes. An introduction part needs to include the current hypotheses regarding the origin and function of GRCs, particularly, the non-recombining nature of GRCs, that may explain few functional genes exist on it.

Minor points:

1. Line 89, here please state clearly what your expectation for GRCs using these two antibodies, i.e., why do you expect to see both on the GRCs in pachytene spread?
2. Line 106, germline genomes, this is unclear, is this testis or ovary? The method part mentioned testes, but would ovary be a better choice?
3. Line 111, how did you identify germline-specific repeats? Also the author need to report the repeat content of GRC and its comparative result to that of the A chromosomes
4. Line 158, this is quite unexpected that the orthologous GRC region between the two genomes is more repetitive. Are they sequence duplications or transposable elements?
5. Line 188, how are these two coverage cutoff determined?

Reviewer #3:

Remarks to the Author:

The manuscript "Rapid gene content turnover on the germline-restricted chromosome in songbirds" by Schlebusch et al presents assembly and analysis of germline restricted chromosomes in nightingale, which have the potential to add substantially to our understanding of the evolution of this intriguing chromosome in the songbird lineage. I'd like to make a few comments that I think will substantially improve the reporting, clarity and interpretability of the work.

1) Many of the discussions in the article center around the presence / absence of genes as suggested by analyses of coverage data that seem to be fairly variable across (unrelated?) individuals even within the same species. As the title and much of the discussion centers in one way or another around the fact that things are observed in one individual / species and not in another one should be careful that non-observations are likely to represent true absence, rather than sampling failure. The best way to do this would seemingly be to perform some sort of validation experiment. For example, it would be very informative to see A) in situ hybridization patterns for the reported chromosome 2 homology region (it would seem that a finch BAC clone or would have a reasonable chance of being a good probe here) and B) PCR validation for at least some of the genes that are reported to vary within and between the two species. Without some sort of orthogonal validation, it is difficult to assess how plausible the presented evolutionary scenarios might be.

2) Information presented in relation to copy number variation in GRC regions would also benefit from a substantially more thorough presentation of mapping depth patterns in relation to somatic reads, especially given that the GRC would seemingly be expected to be present at less than 1/2 model coverage sure to the presence of non-germ cells in the testes and that these proportions could vary among individuals. Reporting the realized coverage of germline (and distribution of values) in relation to A chromosomes for all sequenced samples will aid in interpreting the copy number estimates and improve the overall presentation of these results. This should also be generally useful information for groups that might want to pursue assembly and analysis of other avian GRCs.

3) In reference to line 274: "Most genes identified on the nightingale GRCs were only partially present, however, with approximately 75% of genes having less than a quarter of their coding region present and only 4% having their whole coding region present" and related discussions. These seem to be related to the assessing the probability of functional loss, however it also seems likely that selective pressure, mode of duplication and gene fusions could result in apparent shortening of a gene with the gene still being functional (or more so) in the germline. This may be difficult to assess fully with the current dataset, but perhaps plotting the proportion of the gene retained in relation to their observed transcription in the testes could provide a bit more insight into the potential for their functionality.

4) The discussion around line 305 " where large stretches of DNA can be added and subtracted rapidly, seemingly without consequence." Raises some other interesting matters that might want attention – Overall, the manuscript doesn't seem to do much in the way of assessing consequences per se, perhaps consider rewording? It seems that the data more directly speak to the probability of the chromosome being transmitted to the next generation.

5) This also brings up the related point that GRCs are often transmitted by the oocyte rather than the testes (though not always). Is it worth considering the possibility that within-individual structural maintenance of the GRC in testes? Might the further analysis of depth distributions for testes reads suggested above provide some insight into this possibility? Validation experiments might also provide some insight here.

6) As these analyses rely fairly heavily on an unpublished nightingale genome assembly, is there a plan to release the genome at the time this manuscript is published? Some of the referencing and the Data availability statement suggest this might not be the case, thought that might not be in line with NPG policy. I would also recommend submitting the GRC assemblies and corresponding gene annotations to NCBI in addition to, or instead of, Figshare to facilitate broader access.

Before addressing the individual comments, we provide here a short overview of the major changes made in the revised manuscript. (1) As was requested by reviewer 3, we have added the somatic genome assemblies and their description to this manuscript (Results lines 125-140 [129-144 with track changes] and Methods lines 490-524 [515-549 with track changes]). Adding these new data also clarifies several issues raised by reviewer 1. (2) We have added new transcriptome data from ovaries and testes (Results lines 286-313 [304-331 with track changes] and Methods lines 595-605 [623-633 with track changes]). (3) We expanded the cytogenetic dataset and showed that within-species variation in the GRC size can be observed in one of the species at the cytogenetic level (see Figure 3B). (4) As requested by reviewer 3, using PCR we independently confirmed the presence of a germline specific *cpeb1* gene copy in both species, and a germline specific boundary between chromosome 2 derived sequence and chromosome 6 derived sequence in *L. luscinia*. (5) Finally, many text modifications, additions and edits have been done to clarify and improve the text.

Reviewer #1 (Remarks to the Author):

This study by Schlebusch makes a more complete germline restricted chromosome (GRC) assembly of two closely related species, the common and thrush nightingale. The authors have produced two of the most complete assemblies of the GRC to date, although still highly fragmented relative to autosome assemblies. Nevertheless, this new assembly helps answer questions around this enigmatic chromosome in songbirds. Tissue was sampled from three individuals of each species. The authors use three approaches based around 10x barcoding technology to identify and assemble GRC-specific sequence. They show that GRC gene content differs dramatically between the two species and even within individuals of the same species. Only 4% of GRC genes had their entire coding region present, while the rest were in pseudogenized fragments, suggesting that the GRC is under little selective pressure. The authors identify only one gene, *cpeb1*, which is present in all individuals sampled without copy number variation and contains the complete coding region. Based on comparison with the A-chromosome paralogue in various avian species, they find that GRC *cpeb1* originated before the oscine/suboscine divergence. Given that this corresponds to the earliest estimates of the origin of the GRC, *cpeb1* is likely essential and helping to maintain the GRC in songbirds.

1.1. Overall, we are excited by the study, which presents an amazing amount of GRC reorganization between closely related species and between individuals within a species. The only other example of such reorganization we can think of is the immunoglobulin genes, but even then, here the GRC

reorganization is an entire chromosome. Because of such a remarkable claim, we think the authors need to alert readers of potential technical limitations that could contribute to a higher than expected chromosome reorganization between and within species. No doubt this is a technically challenging chromosome to sequence and assemble to begin with.

Response: In the revised manuscript we have noted the technical limitations of our method which could affect the GRC assembly (lines 150-156 [154-160 with track changes]):

“This method aims to assemble not only GRC sequences that are divergent from the A chromosomes, but also GRC sequences that are similar/identical to A-chromosomal sequences, provided they have the same 10x barcode as a high confidence GRC derived read. It should be noted, however, that any large region without a high enough density of markers identifiably GRC in origin would still be missing or fragmented in the assembly. This is in addition to the normal pitfalls of an Illumina based assembly, such as sequencing biases and an inability to resolve repetitive regions (Peona, et al. 2021).”

1.2 First, the reference somatic assembly is done with short reads, using the 10X genomics protocol. The authors should mention the fragmented limitations of such assemblies, as shown in Rhie et al 2021 Nature for the Vertebrate Genomes Project (VGP).

Response: This was a misunderstanding. The reference somatic genome used is a high-quality genome created using a combination of Nanopore, 10x linked-read sequencing and Omni-C data. We apologize that this wasn't very clear originally, as the description of this genome assembly originally referred to an unpublished paper. We have now moved the description of these assemblies fully into this paper (see the section “Reference somatic genome assembly” lines 125-140 [129-144 with track changes] and the section “Sequencing and assembly of the two nightingales' reference somatic genomes” lines 490-524 [515-549 with track changes]). We have also made it clearer which genome we are referring to in each section. For example, line 550 [575 with track changes] now says “respective germline genome” instead of “respective genome” and line 558 [584 with track changes] now says “aligned to the reference somatic genomes” instead of “aligned to draft somatic genomes”. We hope that the text is now more understandable.

1.3 Second, the authors selected the reads for the GRC assembly by subtraction of those that did not match the somatic reference chromosomes, using three parallel approaches. This unmapped GRC reads by definition are going to result in a more fragmented GRC assembly, because the testis GRC reads that are identical to the somatic chromosomes will also be removed, including proportions of genes. This caveat should be mentioned.

Response: The method used can identify GRC reads that are identical to A-chromosomal sequences due to the 10x linked-read sequencing approach, where the identifiable GRC reads will share barcodes with adjacent regions even if the latter are identical between GRC and regular chromosomes. We have improved the description of our GRC assembly approach in the Results (lines 150-152 [154-157 with track changes]) so that this is clear. But we agree that this is far from perfect, and of the GRC reads that are missed, they will be disproportionately similar to the A-chromosomal sequence. We have mentioned this caveat in the Results (lines 152-156 [157-160 with track changes]).

1.4 It is hard to imagine how can over 90% of the genes loss parts of coding sequences in the GRC. This would have to be due to a highly recombinant process that breaks up and reassembles pieces of somatic DNA into the GRC. An alternative interpretation is that many of the protein coding sequences are missing due to them being identical to the somatic chromosomes and thus subtracted out before assembly of the GRC.

Response: We consider it unlikely that the observed gene fragmentation is the result of persistent assembly errors resulting from an inability to identify GRC sequences that are identical to the A chromosomes. If it was an error, one would predict there to be far more fragmentation in the lower quality assembly of *L. megarhynchos* (see Table 1). One would also predict more gene fragmentation in the GRC that showed lower divergence from the A chromosomes (again *L. megarhynchos*; see Figure 5). Both GRCs would be similarly close in divergence from the A chromosomes (at whatever the limit of one's ability to differentiate the sequences was). One would also predict there to be very little to no fragmentation in the older genes shared with zebra finch (since they have had time to diverge; see Table 2). The genes that are fragmented (including those shared with zebra finch) normally have the same region

present between the two assemblies (see Supplementary Figure 6). And while one could partly explain this away with regions of a gene being conserved or divergent, that argument would then be incompatible with all the genes where a large portion was found in one of the GRCs (sometimes in multiple copies on different scaffolds) but completely absent in the other GRC assembly. And finally, one wouldn't predict the new result we have now added to the revised manuscript where we find that the observed proportion of a gene in the GRC predicts its expression (even after taking the smaller size into account; See Figure 9). We understand that such massive gene degeneration may seem unlikely as it does not normally occur on regular chromosomes. But the GRC only occurs in the germline, which greatly reduces the selective pressure acting on it. What's more, the "unlikely" process described by the reviewer is the sort of thing that would be required in order to get the large variations in GRC size over short evolutionary periods observed using cytogenetic approaches (Torgasheva, et al. 2019; Sotelo-Muñoz, et al. 2022). We have included some more discussion on this topic in the Discussion (lines 376-386 [395-405 with track changes]) to better support our conclusions.

1.5 I think the evidence for different size GRCs between species and individuals is more convincing. But I wonder if another biological explanation could be at play, different developmental stages within an individual. As the primordial germ cells (PGCs) mature, maybe their GRC chromosome is undergoing different stages of germline chromosome reorganization, and what the authors are catching are cells in gonads that are in different stages of reorganization between and within species. If this seems plausible, it should be mentioned as a possible alternative explanation to consider.

Response: We think that the large variation in GRC size among individuals and species is caused by relatively frequent structural changes occurring on this chromosome and low selective pressure acting on it as it is only found in the germline. We do not think that it is likely that the observed variation in GRC size among individuals and species is caused by the reorganization of this chromosome during male germline development as our cytogenetic observations do not support this hypothesis. The GRC does not show marked differences in size among individual germ cells during the pachytene stage in the nightingale species and it seems to be completely eliminated from the nucleus after the pachytene stage. However, we have mentioned in the Discussion (lines 362-365 [380-383 with track changes]) that analysis of the GRC in the female germline would be useful in the future.

1.6 Given that GRC size estimates from the genomic data indicate significant within species variation (Figure 3), we wonder if the authors could identify any similar size variation from the chromosome spreads as well?

Response: In the revised manuscript we expanded our analysis to meiotic spreads from three individuals from each species (10 cells per individual). These meiotic spreads were from different individuals than the genomic data, so it was not possible to directly compare the GRC size estimates from these two sources. However, although we did not observe any significant difference in GRC size in *L. megarhynchos*, we did observe variation in *L. luscinia*'s GRC size among the individuals. See the new version of Figure 3 and Results (lines 114-123 [117-127 with track changes]).

1.7 The descriptions of the GRC assembly in the main text results and methods are confusing. The supplementary Figure 3 version of it is much more understandable. I recommend moving this figure to the main text, and explain the assemblies based on it. For example, the main text has apparent Steps 1), 2), and 3), without any statement that these refer to the pipelines of the left, middle, and right columns of Supplementary Figure 3. We had to figure that out after several readings. Does Somatic genome in the middle column mean the kidney? For the very last step, was Supernova used to assemble these apparent GRC reads? This final assembly step does not seem to be described in the methods?

Response: We have now moved the Figure describing the GRC assembly method to the main text as suggested (Figure 2). We have also added the numbers 1), 2) and 3) to the figure to better link the parts of the figure with the text in the Results and Methods. Additionally (see Response 1.2), we have clarified throughout the text when we are referring to the reference somatic genome assembled with Nanopore, 10x Genomics linked-read, and Hi-C data and when we are using the germline genome assembly.

Please note that the first line of the Method section (lines 543-545 [568-570 with track changes]) does describe the Supernova assembly “Linked 10x reads that originated from the GRC were identified using the following methods before being assembled using Supernova (Weisenfeld, et al. 2017) and the “megabubbles” option (method visualised in Figure 2)”. However, we have also added this information to the new Figure 2 (last box).

1.8 Line 137. The copy number and coverage description needs to be explained better. “Collapsed repeats” is usually the term for artificial collapsing of duplications, shown with increased coverage of raw reads. “Expanded repeats” refers to the calculation that the authors made for estimating to true size of the chromosome if repeats were expanded. But the authors did not actually expand (lengthen) and reassemble the repeats? They give the impression that this was done.

Response: We calculated, based on coverage, an estimated copy number, but this did not require any reassembly. We have modified the text in the revised manuscript in order to not give the impression that we lengthened or reassembled any sequence (lines 187-191 [192-197 with track changes]). We hope the text is more understandable now.

1.9 Line 246. I wouldn't call these GRC assemblies high-quality yet, despite them being the most complete thus far. There are still draft quality assemblies; they can say the most complete draft-quality assemblies. High-quality GRC assemblies will probably require using Pacbio HiFi reads and more advances in technology to isolate the GRC or PGCs from other cells.

Response: We agree. The phrase was meant as a comparative statement to highlight the poor quality of GRC assemblies to date, but it is too easily misunderstood. We have deleted the words “high quality” and line 342 [360 with track changes] now reads “These assemblies thus represent the most complete GRC assemblies achieved to date, although they are still fragmented.”

1.10 Line 269. The Prum et al 2015 bird tree has been shown to be mostly a protein-coding tree that does not reflect the rest of the genome (Jarvis et al 2014; Reddy et al 2017). I think citing Jarvis et al and Reddy et al for the relationship of songbirds with other species in the tree would be more appropriate.

Response: We have changed the reference as suggested (lines 368-370 [386-388 with track changes]).

1.11 lines 301 and 302: There is quite a bit known regarding *cpeb1* in the literature. Can the authors include a little more discussion of that? But, the discussion does have redundancy in it, and they can shorten that redundancy too.

Response: We have added more information about the *cpeb1* gene in the Discussion and removed redundancy (lines 410-421 [431-442 with track changes])

1.12 line 373: Do the authors have any concern with not using a PCR-free DNA-seq kit to reduce coverage biases? Please discuss potential PCR and coverage bias.

Response: There is a 10-cycle PCR step in the 10x Genomics linked-read library preparation protocol which is necessary to add the sequencing primers to the fragmented DNA molecules and PCR is also often used during library preparation for standard Illumina sequencing. Therefore, DNA sequences that are hard to amplify will probably be under-represented in the 10x linked-read assembled sequences. This could have contributed to the fragmented nature of the GRC assemblies and can also affect coverage in certain regions. However, this source of bias should normally affect all the individuals in these regions similarly and should therefore not lead to copy number polymorphism. Despite these limitations, coverage is still often used

to estimate copy number variation. We have shortly mentioned this issue in the Results (Lines 170-173 [175-178 with track changes]) “Although the coverage can vary due to sequencing and PCR bias, as well as just random chance, it can be used to identify regions of the GRC that are duplicated within the GRC but have not diverged sufficiently from their GRC paralogs to be differentiated by the assembly process (Chiang, et al. 2009; Wang, et al. 2014).”

1.13 lines 390-399: Why is a different reference genome used in method 1 and 2? Was there something to be gained?

Response: In method 1, we needed to identify testis-specific sequences occurring in the germline genome, but not in the somatic genome. We thus used an assembly of the germline (testis) genome as a reference. For method 2 on the other hand, we needed to identify testis-specific SNPs with SNP calling methods in sequences occurring in both somatic and germline genomes. It was thus better to use the somatic reference genome. Method 1 targeted highly divergent sequences between somatic and germline genomes while method 2 targeted highly similar sequences. We have slightly modified the text in Methods so that our strategy is better understandable (lines 546 and 555 [571 and 581 with track changes]).

1.14 lines 414-420: This paragraph is unclear and can be rephrased.

Response: The paragraph has been re-worded (lines 580-589 [607-617 with track changes]). We hope it is clearer now.

1.15 Line 439. Blastn is not a good tool for genome alignments. Blastn makes partial alignments. Cactus, Dialign, ClustalW don't delete in between unaligned regions, and thus one gets a more complete alignment.

Response: Given the fragmented quality of the assemblies and the fact that different parts of a scaffold could have different origins, it was thought that a local aligner like Blast would be better. Given the reviewer's concerns however, we compared our results using Blast to results generated with nucmer (a purpose-built genome alignment program which we have experience with). The results agreed 88% of the time for *L. megarhynchos* and 95% of the time in *L. luscinia*. We are happy to use these results instead if they are preferred, but since both methods gave similar results and it isn't obvious which is better, we decided to keep the original blast results.

1.16 Line 458. Here the authors will need to have a public accession number of genebank deposited assemblies.

Response: We have now done this (see Data Availability lines 652-658 [683-689 with track changes]).

1.17 Figure 2 legend needs more information in order for readers to follow what is happening on the Y and X axis.

Response: We have given more detail in the legend of the figure (now Figure 3A).

1.18 Figure 4. Are the chromosome numbers the same as the zebra finch? Further, which zebra finch assembly was used for assigning chromosomes? The Rhie et al 2021 zebra finch assembly is the most updated GCF_003957565.2 for chromosomes. The version used needs to be mentioned and cited.

Response: We have added a new Supplementary Figure 3 in the revised manuscript which shows the homology between the nightingale and zebra finch chromosomes. As can be seen there, we have labelled the nightingale chromosomes the same as the zebra finch's. We have also added information about the version of the zebra finch assembly used (GCA_003957565.2; line 522 [547 with track changes]).

Reviewer #2 (Remarks to the Author):

Schlebusch et al. in this submitted ms investigated two recently diverged songbird species, common and thrush nightingale, for their germline restricted chromosome (GRC) sequences, and compared them between each other, and vs. their regular chromosome (A) sequences for their origins. They extracted the GRC sequences either based on their absence in somatic genome assemblies, or derived from the reads that have diagnostic SNPs or repeats that are specific to the germline. Due to the intrinsic limitation of the sequencing method (Illumina or 10x) and the repetitive nature of GRCs, the assembly is fragmented; but their analyses indicated that there can be large copy number variations of GRC sequences between individuals, and also between species. Although of recent divergence, GRCs of the two species seem to share very few genes and also underwent independent sequence amplification. Few genes annotated in both species' GRCs seem to be intact, with only one gene, CPEB1 that is inferred to be shared by both. There are many recent studies elucidating the origin and evolution of GRCs in songbirds (and absent in other bird species), started from the first genomic study of GRC in zebra finch (Kinsella et al. Nat. Comm.) to more recently, a more systematic inspection of its existence in 16 songbird species (Torgasheva et al. PNAS). I think to gain more genomic insights into GRCs of more species is important for our understanding of their origin and function, if any. However, I have to say, the new insights from this paper, compared to previous work, are not significant enough to merit a publication on Nat. Comm. My major comments are as follows:

2.1. One major conclusion of this work is many GRC-derived sequences contain duplications, inferred only from normalized read coverage data. However, read coverage can be notoriously variable due to the different GC or repeat content from Illumina reads, and this can be indicated by the large variation observed between different scaffolds from GRCs shown in Figure 2. I did notice that, there are higher levels of coverage variations between *L. megarhynchos*' individuals than those of *L. Luscinia*, however additional methods (e.g., check if there are actual reads spanning the duplicated sequences' boundaries, or violations of mate-pair relationships, or any evidence from the assembled sequences)

are needed to make sure the duplicated regions are real. Before that, I would be very cautious to conclude, e.g., at line 141, that the current assembly size is already close to the estimate from meiotic spreads. As both are quite inaccurate that the actual size of the chromosome cannot be accurately scaled by their length, as highly heterochromatic chromosomes tend to be more compact, i.e., shorter than its scaled length.

Response: We agree with the reviewer that read coverage can be affected by factors other than sequence duplication and we have modified the text to reflect this possibility in the revised manuscript (lines 170-173 [175-178 with track changes]). In addition, we noted in the Results (line 198 [205 with track changes]) that the GRC size estimates from both the genomic and cytogenetic data should be viewed with caution. Nevertheless, despite this, the sequence coverage in Illumina data contains relevant information about the variation in sequence copy number as has been shown in multiple studies (Chiang, et al. 2009; Wang, et al. 2014). In addition, we have controlled for the different length of the univalent GRC and bivalent A chromosomes in pachytene (see Materials and Methods lines 480-483 [506-509 with track changes]) and we have stressed it in the revised version of the manuscript (lines 119-122 [123-125 with track changes]).

2.2. The paper reported that at line 183 “585 genes were identified in *L. megarhynchus* GRC, and 406 in another species”. However, it later turned out that “vast majority of these genes are duplicated and only partially present”. So this is quite misleading, as the result seems to be these numbers of genes only means gene fragment, and it is unclear how many of these fragments present on the GRCs actually have an intact open reading frame, or at least have any evidence of transcription. Therefore, it is difficult for the reader to assess whether the GRCs are functional or completely a parasitic element. The authors should provide information regarding the coding potential of these genes as well as the transcription evidence.

Response: We have modified the text to be clearer about the number of identified genes on the GRC (line 241 [254 with track changes]). We have also added a new analysis of the coding potential of GRC genes and provided information on how many GRC genes have or lack a premature stop codon (lines 252-257 [266-270 with track changes]). Finally, we have included

in the revised manuscript an analysis of gene expression data for testis and ovaries (Materials and Methods lines 595-605 [623-633 with track changes]) and provided information on how many GRC genes are expressed in the gonads (see the section “GRC gene expression”; lines 286-313 [304-331 with track changes]).

2.3. Figure 5 is very unclear to me: in the main text, please introduce what is the hypothesis for using this mismatch pattern to determine when GRC originated regarding the two species divergence time. There are no legends in Figure 5, which only shows two density plot of distribution. It is also unclear how the authors estimated “1Mb of the assembly align better...”. I assume that the authors hypothesis sequences that are more similar to one species’ genome could have originated after this species diverged from the other. But how is this pattern relate to Figure 4? As if the GRC sequence originated after the speciation, at least this sequence should have a different A chromosome origin that is absent in the other species’ GRCs, is that so? And how many sequences are like that?

Response: In the revised manuscript, we have explained in the Results (lines 229-233 [242-246 with track changes]) what the expectations of divergence are in the case of a GRC sequence origin before or after the species split: “If the GRC sequence was more similar to the A-chromosomal sequences of its respective species than it is to the other species’ A chromosomes, it suggests that it was derived after the species diverged. By contrast, if the GRC showed similar levels of divergence from the two species’ A chromosomes, it suggests that the GRC origin predates the speciation.” The 1 Mbp number is the area under the curve of Figure 5, but we agree that that is not clear from Figure 5 itself, and as such the reference to Figure 5 has been removed and instead the 1 Mbp number is simply reported as a standalone result. The legend of Figure 5 has been changed so that each line is labelled *L. megarhynchos* A chromosomes and *L. luscinia* A chromosomes, instead of just the species name.

Figure 4 shows that large portions of the GRC have regions that have origins not present in the other species’ GRC. But the presence of these regions alone is not enough to show that they were derived after the speciation event, as the region could have been deleted from one of the species after the speciation event instead. This is the reason why Figure 5 is needed. We hope that our text modifications have made the text clearer.

2.4. It is also unclear that the gene fragment on the GRC is caused by the assembly gap or it is fully assembled, but the sequence open reading frame has been disrupted.

Response: We believe that on the whole, the gene fragmentation is real and not the result of assembly gaps (although the odd exception may occur). We present in the revised manuscript multiple lines of evidence that support this conclusion as justification (see Response 1.4 to reviewer 1, who had a similar comment).

2.5. The discussion about CPEB1 is too short, at least this gene I considered as one major finding of the work, that it seems to be originated at the base of songbirds and could potentially explain the existence of GRC. In the text, the author wrote “CPEB1 is known to play a role in transcript modification during oocyte maturation...this makes it the best candidate for preventing the loss of the GRC from germline”. If I understand it correctly, now songbird germline contain an extra copy of CPEB1 than other birds, and it is also unclear whether the GRC-linked CPEB1 actually perform the same function as the copy on the A chromosome; and even it indeed has the same function, how would this extra copy matter to the “oocyte maturation”? These factors all need to be discussed.

Response: We have expanded the discussion on the *Cpeb1* gene in the revised manuscript as the reviewer suggested (lines 410-423 [431-443 with track changes]). We have also added new evidence of gene expression from the GRC *cpeb1* paralog in ovaries, but not in testis (line 299 [317 with track changes]) and spoken about its divergence (lines 279-283 [297-300 with track changes]).

2.6. There should also be introduction/discussion about the distinction and similarity between the GRC and B chromosomes, both of which are extra non-mendelian chromosome elements in addition to the A chromosomes, and carry mostly repetitive sequences and few functional genes. And introduction

part needs to include the current hypotheses regarding the origin and function of GRCs, particularly, the non-recombining nature of GRCs, that may explain few functional genes exist on it.

Response: We have added a new paragraph about B chromosomes and their differences/similarities to the GRC as well as hypotheses regarding the origin and function of the GRC to the Introduction (lines 58-70 [59-71 with track changes]). Additionally, we mentioned in the Discussion that the absence of effective recombination on the GRC could contribute to rapid gene degeneration (line 385 [404 with track changes]).

2.7. Minor points:

2.7.1. Line 89, here please state clearly what your expectation for GRCs using these two antibodies, i.e., why do you expect to see both on the GRCs in pachytene spread?

Response: We have explained in the revised manuscript how the GRC can be recognized using anti-SYCP3 and CREST antibodies (lines 102-105 [105-108 with track changes]).

2.7.2. Line 106, germline genomes, this is unclear, is this testis or ovary? The method part mentioned testes, but would ovary be a better choice?

Response: We now specify that we used testis (line 144 [148 with track changes]). Males are better to use than females because males have more germ cells in their testis, resulting in better sequencing coverage across the GRC relative to the A chromosomes.

2.7.3. Line 111, how did you identify germline-specific repeats? Also the author need to report the repeat content of GRC and its comparative result to that of the A chromosomes

Response: The methodology for how the germline-specific repeats were identified is given from line 565 to 572 [592 to 599 with track changes]. We have not quantified the repeat content of the GRC compared to the A chromosomes as part of this study as it would ideally need a less fragmented GRC assembly, and it is beyond the scope of this paper.

2.7.4. Line 158, this is quite unexpected that the orthologous GRC region between the two genomes is more repetitive. Are they sequence duplications or transposable elements?

Response: We do not know the answer to this question at the moment and have instead aimed to stay agnostic about the identity and mechanism behind the within-GRC duplications. We hope to generate more contiguous GRC assemblies using long read technology in the future in order to better answer this sort of question.

2.7.5. Line 188, how are these two coverage cutoff determined?

Response: Line 188 (now line 246 [259 with track changes]) does not mention a coverage cutoff. It is reporting the average estimated copy number genes in the two species. This number is based off the number of times a gene was assembled and the estimated copy number of the scaffolds which the gene is on. The method for calculating the coverage is described from line 607 to 620 [635 to 648 with track changes].

Reviewer #3 (Remarks to the Author):

The manuscript “Rapid gene content turnover on the germline-restricted chromosome in songbirds” by Schlebusch1 et al presents assembly and analysis of germline restricted chromosomes in

nightingale, which have the potential to add substantially to our understanding of the evolution of this intriguing chromosome in the songbird lineage. I'd like to make a few comments that I think will substantially improve the reporting, clarity and interpretability of the work.

3.1) Many of the discussions in the article center around the presence / absence of genes as suggested by analyses of coverage data that seem to be fairly variable across (unrelated?) individuals even within the same species. As the title and much of the discussion centers in one way or another around the fact that things are observed in one individual / species and not in another one should be careful that non-observations are likely to represent true absence, rather than sampling failure. The best way to do this would seemingly be to perform some sort of validation experiment. For example, it would be very informative to see A) in situ hybridization patterns for the reported chromosome 2 homology region (it would seem that a finch BAC clone or would have a reasonable chance of being a good probe here) and B) PCR validation for at least some of the genes that are reported to vary within and between the two species. Without some sort of orthogonal validation, it is difficult to assess how plausible the presented evolutionary scenarios might be.

Response: We have created 8 sets of primers targeting 4 regions of interest (3 GRC genes, including *cpeb1*, and a unique boundary created by the duplication from chr2 onto the *L. luscinia* GRC). All 3 GRC genes showed the predicted gonad-specific PCR amplification confirming that our approach to identify GRC sequences was successful. The boundary primers showed amplification only in the *L. luscinia* GRC as predicted, confirming the largest source of sequence difference between the GRCs is real. We have only included the *cpeb1* and boundary PCR results in the paper. The other two genes, while yielding a positive result, were chosen for their low copy number and our ability to design primers to them and they do not represent anything particularly interesting or noteworthy. See Results (lines 220-224 and 283-284 [233-237 and 300-302 with track changes]), Materials and Methods (lines 639-650 [670-681 with track changes]), Supplementary Table 5 and Supplementary Figure 5. The other two genes that weren't included in the paper are reproduced at the end of this document. If needed, we can test more genes.

3.2) Information presented in relation to copy number variation in GRC regions would also benefit from a substantially more thorough presentation of mapping depth patterns in relation to somatic reads,

especially given that the GRC would seemingly be expected to be present at less than ½ model coverage sure to the presence of non-germ cells in the testes and that these proportions could vary among individuals. Reporting the realized coverage of germline (and distribution of values) in relation to A chromosomes for all sequenced samples will aid in interpreting the copy number estimates and improve the overall presentation of these results. This should also be generally useful information for groups that might want to pursue assembly and analysis of other avian GRCs.

Response: We have added information about GRC coverage in relation to A chromosome coverage to the Results (lines 167-170 [171-175 with track changes]): “The single copy GRC coverage for each sample was estimated to be between 14-23% of the A-chromosomal coverage in *L. megarhynchos* and 15-20% in *L. luscinia*. These values were used to normalise coverage values (see Methods) and calculate coverage across the GRC for the three individuals from each species.”

3.3) In reference to line 274: “Most genes identified on the nightingale GRCs were only partially present, however, with approximately 75% of genes having less than a quarter of their coding region present and only 4% having their whole coding region present” and related discussions. These seem to be related to the assessing the probability of functional loss, however it also seems likely that selective pressure, mode of duplication and gene fusions could result in apparent shortening of a gene with the gene still being functional (or more so) in the germline. This may be difficult to assess fully with the current dataset, but perhaps plotting the proportion of the gene retained in relation to their observed transcription in the testes could provide a bit more insight into the potential for their functionality.

Response: We have calculated the relationship between the completeness of a gene and its expression, as suggested by the reviewer, and it was indeed very interesting. We now present the data in Figure 9 and Supplementary Figure 9 and mention it on lines 305-313 [323-331 with track changes]: “Interestingly, the amount of gene expression was also correlated (exponentially) with the proportion of that gene found on the GRC (even after correcting for the smaller size of the gene fragment), with more complete genes showing higher expression than fragmented genes (see Figure 9).” As we mention in the Discussion (line 382 [400 with

track changes]), we believe that the relatively low expression from short fragmented GRC genes could be incidental and non-functional.

Nevertheless, we also briefly discussed the possibility that some shortened genes could acquire a novel function (lines 376-379 [395-398 with track changes]) “While the occasional isolated gene exon or exons may acquire a novel function, for example as a long non-coding RNA (Ponting, et al. 2009), it is difficult to imagine that such functional gene shortening would occur simultaneously across hundreds of genes in such a short period of time. “

Finally, we have added a new analysis of coding potential of GRC genes in the revised manuscript (see Results line 252-257 [266-270 with track changes]).

3.4) The discussion around line 305 “where large stretches of DNA can be added and subtracted rapidly, seemingly without consequence.” Raises some other interesting matters that might want attention – Overall, the manuscript doesn't seem to do much in the way of assessing consequences per se, perhaps consider rewording? It seems that the data more directly speak to the probability of the chromosome being transmitted to the next generation.

Response: We agree. We have reworded the sentence (lines 424-426 [445-447 with track changes]) to say “... where large stretches of DNA can be added and subtracted rapidly, seemingly without consequence for the fitness of the organism and the probability of the chromosome being transmitted to the next generation.”

3.5) This also brings up the related point that GRCs are often transmitted by the oocyte rather than the testes (though not always). Is it worth considering the possibility that within-individual structural maintenance of the GRC in testes? Might the further analysis of depth distributions for testes reads suggested above provide some insight into this possibility? Validation experiments might also provide some insight here.

Response: Our cytogenetic observations do not support the hypothesis of within-individual large structural changes on the GRC. The GRC in the pachytene stage does not show large differences in size among individual germ cells and it seems to be completely eliminated from the nucleus after the pachytene stage. However, to acknowledge the theoretical possibility that the GRC could show some differences in the male and female germline, we have added the following sentence to the Discussion (line 362-365 [380-383 with track changes]) “We should, however, note that our data does come from the male germline, where the chromosome is not normally heritable. It would be good in the future to confirm that there are no sequence differences between a heritable GRC in the ovary and a non-heritable GRC in the testis.”

With regards to the potential for additional depth calculations, it is unclear what we would be looking at. We would like to say that the results presented suggest that this is not the case, but realistically there are endless possible ways in which one could imagine such processes occurring and they would all produce different results. It is thus unclear what would count as a positive result versus a negative result.

3.6) As these analyses rely fairly heavily on an unpublished nightingale genome assembly, is there a plan to release the genome at the time this manuscript is published? Some of the referencing and the Data availability statement suggest this might not be the case, though that might not be in line with NPG policy. I would also recommend submitting the GRC assemblies and corresponding gene annotations to NCBI in addition to, or instead of, Figshare to facilitate broader access.

Response: We have added the genome assemblies to this paper now. We describe their assembly in the section “Sequencing and assembly of the two nightingales’ reference somatic genomes” on lines 490-524 [515-549 with track changes] and present the results in the section “Reference somatic genome assembly” on lines 125-140 [129-144 with track changes].

PCR products for GRC validation visualized on 1% agarose gel for reviewer 3, response 3.1. The primer pairs used were as follows: XP_005046862_p1_F and XP_005046862_p1_R (A), XP_005046862_p2_F and XP_005046862_p2_R (B), XP_016160863_p1_F and XP_016160863_p1_R (C), and XP_016160863_p2_F and XP_016160863_p2_R (D). The lanes correspond to 100bp New England Biolabs ladder (L), *L. megarhynchos* testis DNA sample (1), *L. megarhynchos* kidney DNA sample (2), *L. lusciniina* testis DNA sample (3), *L. lusciniina* kidney DNA sample (4), and negative control (N).

Reviewers' Comments:

Reviewer #2:

Remarks to the Author:

Dear Editor,

Sorry for the late review,

1. For the main comments, the authors did not address the novelty issue that I raised, so what new insights have this work gained compared to Torgasheva et al. PNAS and other previous works?
2. For questions 1, the authors acknowledged the issue of using read coverage to infer duplications, they still did not provide any additional evidence of duplications on GRC as I suggested in the last round of review. Again, are there any reads spanning the duplication boundaries, or read pairs with violated mate-pair relationship that indicate the duplications on GRC?
3. Figure 5 is still misleading: the authors showed that the GRC sequences have a similar distribution of % of nucleotide mismatches to one species but not the other, then conclude the GRC originated after the speciation (if I understand correctly). This does not necessarily mean the GRC has more similar sequences to one species, just means it has more similar levels of mutation rate to one species. The direct evidence should come from the alignment between the GRC sequences vs. the A chromosomes of the two species. In the text line 234 "with approximately 1Mb of the assembly aligning better..", but no results are shown for this sentence.

Reviewer #3:

Remarks to the Author:

I appreciate the changes that were made in reference to my and the other reviewer's comments. I would like to make a few additional comments that I think, if addressed, will improve the presentation of the manuscript and make it suitable for publication.

- 1) Regarding presentation of the assembly qualities, given that their estimates of completeness range from 36-75% it probably isn't appropriate or useful to use the word "majority ... for both species" perhaps "a large fraction ... for both species". Moreover, the assessments of completeness could use a bit more depth of presentation, including N50/L50 scaffold and contig stats.
- 2) Some additional information and comparisons against somatic component of the 10X-based assembly would provide vital context. What is the degree of completeness and gene model fragmentation in the 10X somatic chromosomes? For a fairer comparison would it not be more appropriate to down-sample 10X barcodes to 15 - 20 % and assess what that does to somatic chromosome assembly.
- 3) Related to this, I think it would be useful to show coverage histograms for somatic and GRC from the 10X data. This would provide a better sense of the underlying means and variation in coverage which are important to assessing many of the assertions made in the paper.
- 4) Figure 5 is hard to interpret in relation to in-text statements about divergence times. I think adding analogous plots showing the degree of divergence between species, probably separately for micro- and macro- chromosomes will provide essential context.
- 5) It is good to see that they performed PCR validation of the CHR2 duplication, performing similar PCRs for segments that are thought to be variable among *L. luscinia* individuals would seemingly go a very long way toward addressing the probability of false negative detections.
- 6) The statement "Chromosome 1 and chromosome 2 were both represented by two scaffolds in the

genome assemblies of both *L. megarhynchos* and *L. luscinia*. This may suggest two chromosomal fission events in the nightingale karyotype". Are there aspects of the karyotype or comparative maps that would support this assertion? Without more information this sentence seems more confusing than useful. Given that the authors performed length measurements of chromosomes, maybe the supporting information exists in data already collected. Would the proposed fission products be expected to be in the size range of macrochromosomes?

Reviewer #4:
None

We appreciate the constructive comments and suggestions and have revised the manuscript accordingly. The major changes are summarised below, followed by a response to individual questions and comments.

1) After evaluating a control suggested by Reviewer 3, we have decided to remove genes from the analysis that don't have at least 25% of their coding region present in at least one species' GRC. This was done to reduce the potential of misidentification of short coding regions falsely inflating the number of genes on the GRCs and the number of differences between the GRCs. However, the removal of these short genes has also removed noise and/or bias from several of the analyses, and as such has improved and/or simplified many findings. For example, the average copy number of genes is now higher, a higher proportion of genes have a premature stop codon present, Figure 9 now shows the exciting tissue specific correlation without the need of the old Supplementary Figure 9 (now removed) and Supplementary Figure 8 now shows correlation within tissues and between species, but not within species and between tissues.

2) Following PCR results which showed that the GRC assemblies were more prone to gaps than we previously thought, we have tried to add degrees of confidence to whether or not a gene or sequence is shared between the two GRCs or not by identifying the potential for shared homology even when sequences don't directly align.

3) Figure 5 was cited by both reviewers as requiring work. We have attempted to improve it by showing the difference in divergence between the two species' A chromosomes rather than just the distribution of said divergence. Subtracting the one species from the other gives the figure much better resolution and hopefully therefore makes it more definitive. We have also done the changes suggested by Reviewer 3 using the original methodology for comparison.

4) Since more alignments were necessary for the revision, we have now done all alignments which don't involve proteins with minimap2. This makes the paper more internally consistent as well as addressing a concern previously raised by Reviewer 1.

Together, these changes along with smaller adjustments have resulted in many improvements to the text and Figures.

Reviewer 2

2.1 “For the main comments, the authors did not address the novelty issue that I raised, so what new insights have this work gained compared to Torgasheva et al. PNAS and other previous works?”

Torgasheva et al. PNAS characterized the GRC size in multiple songbirds, but only did so cytogenetically. They show that the GRC size varies greatly between species but were not able to determine to what degree the GRCs differed in genes and genetic content in general. There is a paper describing genetic content of the GRC in zebra finch (Kinsella et al. 2019), but they only assembled a small part of the macro GRC (approximately 1%). Another paper, which was submitted after this one, describes the GRC genetic content in the blue tit (Mueller et al. 2023).

We are the first to compare gene content between species, which allows us to study novel evolutionary dynamics of the genetic content on this chromosome. We show that large differences in genetic content are present even in situations with closely related GRCs that

look similar sizes under the microscope, suggesting rapid gene content turnover on the GRC. This might have implications for songbird speciation. We are also the first paper to suggest that a large proportion of genes are non-functional. And, while the evidence is preliminary, we are the first paper to suggest that the RNA expression in the testis is effectively random. This is likely a consequence of the GRC only being present in germ cells and not regularly inherited through males, leading to relaxed selection on this chromosome. Finally, we performed the first comparative analysis of GRC genes with the aim of revealing conserved and functionally important genes which might explain the conservation of the GRC for 47 million years in the songbird lineage. Our analysis revealed only one gene with a complete coding sequence shared between the two nightingales which did not show copy number variation. This gene represents a good candidate for the functional indispensability of the GRC. We have further stressed the novelty of our paper in the Introduction (line 88 [89 with track changes]) and Discussion (lines 385-387, 414-416, 506-508 [lines 413-415, 449-451, 469-470 with track changes]).

2.2 “For questions 1, the authors acknowledged the issue of using read coverage to infer duplications, they still did not provide any additional evidence of duplications on GRC as I suggested in the last round of review. Again, are there any reads spanning the duplication boundaries, or read pairs with violated mate-pair relationship that indicate the duplications on GRC?”

The GRC assemblies are highly fragmented and chaotic. This at least partially appears to be a result of many internal duplications, large and small. A sequence may be duplicated many times in the GRC and some of these duplications may be tandem and others not. Depending on their varying degrees of sequence similarity, some of these duplications may be differentiated in the assembly and present as multiple scaffolds (see Supplementary Figure 4) while others may not be.

We expect the assembler to break up contigs and scaffolds at the point where the correct adjoining sequence becomes ambiguous. This will often result in the boundaries of a duplicated sequence either being at the ends of the scaffold, unassembled or present in another short contig. This means that we do not have a good idea of where duplication boundaries lie. So, while we have many read pair violations across the assembly, it is unclear how the suggested analyses would work in these conditions.

To try and address the reviewer’s concerns, we have looked for haplotypes on scaffolds (see attached Figure A at end of the document). Since the GRC is a single copy chromosome in males (as sequenced here), each haplotype should represent an undifferentiated duplication. And as can be seen, there are many such examples of these haplotypes and they anecdotally appear to correlate with the estimated copy number from coverage.

In addition, we have added the standard deviation associated with our gene copy number estimates to Figure 7 and Supplementary Figure 6. While this can’t account for coverage shortcomings, such as sequencing bias, it does convey to the reader how variable the coverage is on the scaffolds the genes come from.

2.3 “Figure 5 is still misleading: the authors showed that the GRC sequences have a similar distribution of % of nucleotide mismatches to one species but not the other, then conclude the GRC originated after the speciation (if I understand correctly). This does not necessarily mean the GRC has more similar sequences to one species, just means it has more similar levels of mutation rate to one

species. The direct evidence should come from the alignment between the GRC sequences vs. the A chromosomes of the two species. In the text line 234 “with approximately 1Mb of the assembly aligning better..”, but no results are shown for this sentence.”

We have improved Figure 5 and hope it is clearer now. If the majority of the GRC sequence predates the speciation of the two nightingales, then we would expect this new distribution to be symmetrical. The more asymmetrical the distribution is, the more the GRC resembles one of the two A-chromosomal sets. The rate of mutation of the GRC should not alter this pattern. A higher GRC mutation rate would (along with other factors) increase the variance around the center but do so in a symmetrical way.

We agree that a very different A-chromosomal mutation rate between the two species could confound this analysis, but we are not concerned by this since A) this seems unlikely for two species that are so closely related with very similar body size, generation time and lifestyles and B) if this was the case, we would see the exact same asymmetrical pattern when aligning the other GRC to the two A-chromosomal sets.

The “1 Mbp” number is derived from the figure. It used to be (in the old Figure 5) calculated from the area under the curve and it is now the sum of the left side minus the sum of the right side. We had previously referenced the Figure when stating the number, but decided it was optimistic to expect the reader to calculate the area under the curve and as such we now state the number in the paragraph talking about the figure without the direct reference. But the number, along with the Figure (old and new), are derived from the alignment of the GRC to the A chromosomes of the two species.

Reviewer 3

3.1 “Regarding presentation of the assembly qualities, given that their estimates of completeness range from 36-75% it probably isn’t appropriate or useful to use the word “majority ... for both species” perhaps “a large fraction ... for both species”. Moreover, the assessments of completeness could use a bit more depth of presentation, including N50/L50 scaffold and contig stats.”

We have changed “majority” to “large fraction” or “large proportion” (lines 96, 205, 353 [lines 97, 206, 376 with track changes]).

The scaffold N50 and L50 have been added to the text (lines 164-166 [lines 166-167 with track changes]). The Scaffold and contig L50s have been added to Table 1 (N50 values were already there).

3.2 “Some additional information and comparisons against somatic component of the 10X-based assembly would provide vital context. What is the degree of completeness and gene model fragmentation in the 10X somatic chromosomes? For a fairer comparison would it not be more

appropriate to down-sample 10X barcodes to 15 – 20 % and assess what that does to somatic chromosome assembly.”

We agreed with the reviewer that this was a good idea for a control. However, we were unsure how the size of the somatic genome (1 Gbp) would affect the comparison. So, to account for size, we made three “artificial GRCs” for each species by selecting regions at random from the somatic genomes that were approximately the same size as that species’ GRC assembly (4 Mbp in *L. megarhynchos* and 6 Mbp in *L. luscinia*). We then selected all 10x somatic reads that aligned to these regions and down sampled them to 16x and 12x coverage for the two species, respectively. This coverage represents the estimated coverage that was achieved for single copy areas by the isolated GRC reads used for assembly. In theory, this method isn’t ideal since there are less reads per barcode than you’d find with just shallow sequencing. But we decided this wasn’t too big of a problem. And sure enough, all the assemblies were still better than the GRC assemblies (see Table A below).

Table A: “Artificial GRC” assembly statistics and the actual GRC assembly statistics for comparison.

	“Artificial GRC” from somatic genome						Real GRC assemblies	
	LM_1	LM_2	LM_3	LL_1	LL_2	LL_3	LMgrc	LLgrc
Total Length	4.0 Mbp	4.0 Mbp	4.1 Mbp	5.8 Mbp	5.9 Mbp	6.2 Mbp	3.5 Mbp	5.6 Mbp
Large Scaf	1.0 Mbp	450 kbp	730 kbp	360 kbp	1.9 Mbp	720 kbp	110 kbp	370 kbp
Num of Scaf	72	154	40	242	213	249	1400	750
Scaf N50	440 kbp	150 kbp	380 kbp	110 kbp	140 kbp	150 kbp	2.8 kbp	46 kbp
Contig N50	71 kbp	46 kbp	79 kbp	43 kbp	64 kbp	48 kbp	2.5 kbp	24 kbp
%N	1.4%	2.6%	0.83%	1.5%	1.2%	2.3%	3.3%	6.4%

This analysis allowed us to compare the gene content in our 6 artificial assembled regions to the contiguous region on the genome from which they originated (effectively controlling for fragmentation). The first thing that became apparent was that short coding sequences were not being identified with accuracy, with 32% of identified genes with less than 25% of their coding region present not being found in the contiguous portion of the genome. This suggests that while the alignment may not be random (low e-value), we do not have the power in these short sequences to differentiate between similar genes. As a result of this finding, we have excluded genes that didn’t have more 25% of their coding region present in at least one of the GRCs from our analysis. This was done to prevent the over estimation of gene numbers on the GRC and the possible exaggeration in the number of different genes between the two GRCs.

While this insight was very important, the analysis was less effective at fulfilling its main function, given the assemblies were of better quality than the GRC assemblies. The gene fragmentation that was observed was also observed on the original contiguous genomic

regions, suggesting it was the result of a lack of conservation between *F. albicollis* and the nightingales, rather than errors due to assembly problems (see Figure B). So, unfortunately the control was inconclusive on the point of assembly gaps causing fragmentation.

Figure B: Comparison of the proportion of genes found between the contiguous genomic regions, the artificial somatic assemblies and the two GRC assemblies. The three contiguous genomic regions are on the left of each graph (labelled with a G, for example LM1G). The three artificial assembled regions are labelled LM1, LM2, etc. The two GRCs are on the right (LM GRC and LL GRC). The blue bar represents the number of genes with 25%-50% of their coding region present, the red bar is 50%-75%, etc.

It should be noted at this point, given the new findings in 3.5, that we have toned down the confidence with which we report fragmentation throughout the paper, but we do not think this has changed the overall conclusions of the paper.

3.3 “Related to this, I think it would be useful to show coverage histograms for somatic and GRC from the 10X data. This would provide a better sense of the underlying means and variation in coverage which are important to assessing many of the assertions made in the paper.”

The coverage distributions for all samples are now provided as Supplementary Figure 9.

3.4 “Figure 5 is hard to interpret in relation to in-text statements about divergence times. I think adding analogous plots showing the degree of divergence between species, probably separately for micro- and macro- chromosomes will provide essential context.”

We have attempted to improve Figure 5. We hope it is more intuitive and informative now. See also 2.3.

The old methodology for generating Figure 5 is provided below (Figure C) with Reviewer 3’s suggested additions. It has been updated to use the Minimap2 alignments rather than the Blast alignments to make it more comparable to the macro and micro chromosome alignments which were suggested. The figure therefore looks different to the old Figure 5, but the information is effectively the same (there are less minimap2 alignments since each alignment

is longer, which means that the percentage of mismatches has increased and bin sizes needed to be adjusted).

Figure C: Distribution of mismatches which result from aligning the GRCs to the two species A chromosomes (left) and the *L. megarhynchus* macro and micro chromosomes to the *L. luscinia* genome (right). The *L. megarhynchus* GRC (top left) has less mismatches to the the *L. megarhynchus* A chromosomes than it does to the *L. luscinia* A chromosomes. Note that the two graphs on the right only have red lines as they represent *L. megarhynchus* A-chromosomal sequences and were therefore only aligned to the *L. luscinia* genome.

3.5 “It is good to see that they performed PCR validation of the CHR2 duplication, performing similar PCRs for segments that are thought to be variable among *L. luscinia* individuals would seemingly go a very long way toward addressing the probability of false negative detections.”

We designed several primers to 3 GRC regions of interest. The first region was a gene (XP_005051812) that was present in the *L. megarhynchus* GRC, but not found in the *L. luscinia* GRC. The other two regions of interest were genes that were present in both species, but shorter in *L. luscinia* than in *L. megarhynchus*. All three loci successfully amplified in the testis sample and not the kidney, confirming their GRC origin. But to our surprise, all three of these PCR reactions showed a positive result in the *L. luscinia* GRC (see Supplementary Figure 5F-L).

These results required a large reworking of the paper. In order to account for the fact that gaps are more common than we once thought, we now no longer just look at whether a sequence aligned to the other GRC (homologous regions), but we also look at whether its mapping position on the somatic genome is close to a sequence from the other GRC (potentially homologous regions). The logic is that it now seems more likely that that piece of DNA was duplicated onto the GRC ancestrally once and the lack of alignment between the two GRCs represents a gap (this accounts for all three of the PCRs in question). In comparison, if no part

of the other GRC aligns anywhere near the presumed genomic origin of a GRC sequence, we conclude that it is species specific (see lines 210-219, 618-620 [lines 212-222, 656-658 with track changes]). Since all genes with any doubt associated with them are classified as potentially homologous, this results in a large maximum overlap and a conservative minimum estimate for differences. Importantly, our main conclusions remain unchanged after this correction.

3.6 “The statement “Chromosome 1 and chromosome 2 were both represented by two scaffolds in the genome assemblies of both *L. megarhynchos* and *L. luscinia*. This may suggest two chromosomal fission events in the nightingale karyotype”. Are there aspects of the karyotype or comparative maps that would support this assertion? Without more information this sentence seems more confusing than useful. Given that the authors performed length measurements of chromosomes, maybe the supporting information exists in data already collected. Would the proposed fission products be expected to be in the size range of macrochromosomes?”

We agree that the phrasing is awkward, but we feel like it should be addressed somehow, even though it doesn't impact the main conclusions of the paper. Unfortunately, it is not clear from the length measurements what the correct answer is.

We have changed the phrasing to: “Chromosome 1 and chromosome 2 were both represented by two scaffolds in the genome assemblies of both *L. megarhynchos* and *L. luscinia*. This may suggest ancestral chromosomal fission events in these two chromosomes” (line 139 [line 140 with track changes]).

Figure A: GRC specific haplotypes indicating undifferentiated duplications in the assembly. Haplotypes were chosen that were present in multiple reads in multiple individuals. Screenshots from IGV (v2.16.0)

GRC_2996: Estimated coverage 2.1x

GRC_3345 : Estimated coverage 3.3x

Reviewers' Comments:

Reviewer #3:

Remarks to the Author:

The new draft of the manuscript "Rapid gene content turnover on the germline-restricted chromosome in songbirds" by Schlebusch et al has improved substantially though the authors' careful consideration of reviewer comments. Overall the paper is much easier to interpret and I think is in pretty good shape. I do have some minor issues with some of the interpretations as they relate to selection and duplication that I outline below. I think the authors should take these into consideration, but I wouldn't want to make changes compulsory since these are more related to the discussion and in a sense show the authors thought process as they proceeded through the described analyses.

1) Several sentences related to duplication as it relates to function or evidence for the degree of selective pressure seem to make logical leaps that I think are not really founded. In this context the authors should recognize that there are many multicopy and copy number variant genes in human that are well understood to have important functions. Studies of the human Y have also shown that the presence of duplicates is important to the maintenance of coding genes on non-recombining chromosomes.

e.g.

"This means that while these genes are likely of ancient GRC linkage, they are still actively undergoing within-GRC duplication and deletion, suggesting little selective pressure acting on them." AND ..."which despite being present in all three species, also show high variation in copy number within species. It thus seems unlikely that they represent indispensable GRC sequences."

2) Similarly, the sentence "This pseudogenisation might be the result of little selective pressure acting on the GRC due to its presence solely in the germline, allowing for the accumulation of structural changes which would be deleterious on any other chromosome." Seems to lack some level of detail and/or logic. Why is it that the authors think that the chromosome's presence in the only tissue that gives rise to the next generation is consistent with the idea that it might be under relaxed selective pressure?

3) Finally, the statement "it suggests that the GRC is largely non-functional". I am not sure any of the data presented here actually suggest that this is true. It seems like a stretch to equate higher rates of gene turnover to the functional relevance of the chromosome.

Response to reviewers #3

We thank the reviewer for their kind general comments. We have reproduced the three specific comments and our responses below.

1) Several sentences related to duplication as it relates to function or evidence for the degree of selective pressure seem to make logical leaps that I think are not really founded. In this context the authors should recognize that there are many multicopy and copy number variant genes in human that are well understood to have important functions. Studies of the human Y have also shown that the presence of duplicates is important to the maintenance of coding genes on non-recombining chromosomes.

e.g.

“This means that while these genes are likely of ancient GRC linkage, they are still actively undergoing within-GRC duplication and deletion, suggesting little selective pressure acting on them.” AND ...“which despite being present in all three species, also show high variation in copy number within species. It thus seems unlikely that they represent indispensable GRC sequences.”

We have changed “This means that while these genes are likely of ancient GRC linkage, they are still actively undergoing within-GRC duplication and deletion, suggesting little selective pressure acting on them.” to say “... which could suggest little selective pressure...”.

We agree that multiple duplications and variation in copy number don't necessarily mean that the genes are not functional. But we believe that our comments don't rule out that possibility. At the same time, we do think that it is more likely that genes that seem more tightly regulated, with little variation within and between species will be the vital genes that we are interested in.

2) Similarly, the sentence “This pseudogenisation might be the result of little selective pressure acting on the GRC due to its presence solely in the germline, allowing for the accumulation of structural changes which would be deleterious on any other chromosome.” Seems to lack some level of detail and/or logic. Why is it that the authors think that the chromosome's presence in the only tissue that gives rise to the next generation is consistent with the idea that it might be under relaxed selective pressure?

We believe that genes on the GRC are under less selective pressure than genes on the A chromosomes because any gene that has an important function in a tissue other than the germline won't ever come into contact with that tissue and therefore won't be able to cause the deleterious effects that its mutations would result in.

We have tried to clarify this point by adding onto the quoted sentence the following: "... where such mutated genes could be expressed in the tissue in which they normally act."

3) Finally, the statement "it suggests that the GRC is largely non-functional". I am not sure any of the data presented here actually suggest that this is true. It seems like a stretch to equate higher rates of gene turnover to the functional relevance of the chromosome.

We agree that the sentence was misworded/unclear. We did not mean that the GRC didn't have a function, but rather that large portions of the GRC did not have a function. We agree that the GRC probably has an important/essential function for the songbirds.

We have changed the text to the following: "...it suggests that large portions of the GRC are effectively non-functional and that the GRC frequently acquires (and subsequently loses) sequences from the A chromosomes."